

# Measurement report: Stoichiometry of dissolved iron and aluminum as an indicator of the factors controlling the fractional solubility of aerosol iron: Results of the annual observations of size-fractionated aerosol particles in Japan

Kohei Sakata[1*], Aya Sakaguchi[2], Yoshiaki Yamakawa[3], Chihiro Miyamoto[3], Minako Kurisu[4], Yoshio Takahashi[3]

[1]Earth System Division, National Institute for Environmental Studies, 16-2 Onogawa, Tsukuba, Ibaraki 305-8506, Japan

[2]Faculty of Pure and Applied Science, University of Tsukuba, 1-1-1 Tennodai, Tsukuba, Ibaraki 305-8571, Japan

[3]Graduate School of Science, the University of Tokyo, 7-3-1 Hongo, Bunkyo-ku, Tokyo, 113-0033, Japan

[4]Research Institute for Marine Resources Utilization, Japan Agency for Marine-Earth Science and Technology, 2-15

Natsuhshima-cho, Yokosuka, Kanagawa 237-0061, Japan

*Correspondence to*: Kohei Sakata (sakata.kohei@nies.go.jp)





**Abstract.**

Atmospheric deposition of iron (Fe) in aerosol particles is enhanced primary production on the ocean surface, resulting in promoting the uptake of carbon dioxide into the surface seawater. Atmospheric deposition of iron (Fe) promotes primary production in the surface ocean, resulting in enhanced uptake of carbon dioxide into surface seawater. Since microorganisms in seawater utilize dissolved Fe (d-Fe) as a nutrient, the bioavailability of Fe in aerosol particles depends on its solubility. However, factors controlling fractional Fe solubility ($Fe_{sol}\%$) in aerosol particles have not been fully understood. This study

performed annual observations of $Fe_{sol}\%$ in size-fractionated (seven fractions) aerosol particles at Higashi-Hiroshima, Japan. In particular, the feasibility of the molar concentration ratio of d-Fe relative to dissolved Al ([d-Fe]/[d-Al]) as an indicator of the sources of d-Fe in aerosol particles because this ratio is likely dependent on the emission sources of Fe (e.g., mineral dust, fly ash, and anthropogenic Fe oxides) and their dissolution processes (proton-promoted and ligand-promoted dissolutions). Approximately 70 % of total Fe and dissolved Fe was present in coarse and fine aerosol particles, respectively, and the average

$Fe_{sol}\%$ in fine aerosol particles (11.4 ± 6.97%) was higher than that of coarse aerosol particles (2.19 ± 2.27 %). In addition, the average ratio of [d-Fe]/[d-Al] in coarse aerosol particles (0.408 ± 0.168) was lower than that in fine aerosol particles (1.15 ± 0.803). The range of [d-Fe]/[d-Al] ratios in the coarse aerosol particles (0.121–0.927) was similar to that obtained by proton-promoted dissolutions of mineral dust (0.1–1.0), indicating that d-Fe in coarse aerosol particles were derived from mineral dust. The [d-Fe]/[d-Al] ratios of aerosol particles ranged from 0.386 to 4.67, and [d-Fe]/[d-Al] ratios greater than 1.5 cannot

be explained by proton-promoted dissolution and ligand-promoted dissolution (1.0 < [d-Fe]/[d-Al] < 1.5). The [d-Fe]/[d-Al] ratio correlated with the enrichment factor of Fe in fine aerosol particles (r: 0.505), indicating that anthropogenic Fe with a high [d-Fe]/[d-Al] ratio was the source of d-Fe in fine aerosol particles. The high [d-Fe]/[d-Al] ratio was attributed to Fe-oxides emitted from high-temperature combustions (high-temp-FeOx). Finally, the fraction of high-temp-FeOx to d-Fe in total suspended particulate (TSP) was calculated based on the [d-Fe]/[d-Al] ratio of aerosols and their emission source samples. As

a result, the fraction of high-temp-FeOx to d-Fe in TSP varied from 1.48 to 80.7 %. The high fraction was found in summer when air masses originated from industrial regions in Japan. By contrast, approximately 10 % of d-Fe in the TSP samples collected in spring and during Asian dust events was derived from high-temp-FeOx, when air masses were frequently transported from East Asia to the Pacific Ocean. Thus, mineral dust is the dominant source of d-Fe in Asian outflow to the Pacific Ocean.



# 1. Introduction

Primary production in high-nutrient, low-chlorophyll regions, such as the North Pacific, Eastern Equatorial Pacific, and Southern Ocean, is limited by the depletion of dissolved Fe (d-Fe, Martin and Fitzwater, 1988; Boyd et al., 2007; Moore et al., 2013; Tagliabue et al., 2017). The atmospheric deposition of Fe activates primary productions in surface seawater, enhancing the oceanic uptake of atmospheric $CO_2$ (Martin, 1990; Martin et al., 1994; Falkowski et al., 2000; Jickells et al., 2005). In the last glacial–interglacial period, atmospheric $CO_2$ concentration was inversely correlated with the supply of mineral dust to the Southern Ocean (Martínez-Garcia et al., 2009, 2011, 2014). Thus, the fertilization of d-Fe in surface seawater via aerosol deposition is an important driver of the global climate system. Given that phytoplankton in surface seawater utilizes d-Fe as nutrients, the bioavailability of Fe in aerosol particles depends on their solubility (Moore et al., 2013). Iron in aerosols is not highly water soluble, and the fractional Fe solubility ($Fe_{sol}$% = [dissolved Fe/total Fe] × 100) in marine aerosols ranges from 0.1 % to 90 % (Sholkovitz et al., 2012; Mahowald et al., 2018). Differences in $Fe_{sol}$% among emission sources (e.g., mineral dust vs. anthropogenic aerosol) and the atmospheric processes of Fe-bearing particles are factors controlling $Fe_{sol}$% (Sedwick et al., 2007; Sholkovitz et al., 2009; Mahowald et al., 2018; Ito et al., 2019, 2021). In fact, fine aerosol particles have been found to yield higher $Fe_{sol}$% than coarse aerosol particles derived from either or both anthropogenic Fe and atmospheric processes, as a result of size-fractionated samplings of marine aerosols (Buck et al., 2010a; Chance et al., 2015; Sakata et al., 2018; Kurisu et al., 2021; Baker and Jickells, 2006, 2017; Baker et al.,2020; Gao et al., 2019).

The source apportionment of dissolved Fe in aerosol particles has been conducted on the basis of the correlation analysis of $Fe_{sol}$% with the concentrations and enrichment factors (EFs) of coexisting elements. For example, the EFs of V and Pb are used as tracer elements of heavy-oil and coal combustion processes, respectively (Sholkovitz et al., 2009; Conway et al., 2019; Hsieh et al., 2022). Indeed, the correlations of $Fe_{sol}$% with the EFs of V and Pb are useful for evaluating the presence of anthropogenic Fe. However, quantitative evaluating the fraction of anthropogenic Fe in d-Fe in aerosol particles is difficult. Therefore, an indicator that can estimate the fraction of anthropogenic Fe in d-Fe in aerosols is required to evaluate quantitatively the Fe supply from aerosols to the ocean surface. The fraction of anthropogenic Fe has recently been estimated by using the Fe isotope ratio ($\delta^{56}Fe$) because anthropogenic Fe has a lower $\delta^{56}Fe$ than mineral dust (Kurisu et al., 2016a and b, 2019, 2021; Conway et al., 2019). Although the $\delta^{56}Fe$ of total Fe in marine aerosol particles has been reported by previous studies, data on the $\delta^{56}Fe$ of d-Fe in marine aerosol particles are limited due to analytical difficulties, including high filter blanks (Conway et al., 2019; Kurisu et al., 2021). Considering that the fractions of anthropogenic Fe in d-Fe depends on the $Fe_{sol\%}$ of anthropogenic Fe and mineral dust, the contribution of non-crustal Fe in total Fe is not always directly reflected in that of d-Fe. Therefore, the ability to produce indicators for the source estimation of d-Fe in aerosol particles with low analytical difficulty is ideal.

The stoichiometry of the mineral dissolution of major elements (e.g., Si, Al, Fe, and Mg) has been investigated to understand the dissolution processes of minerals in the environment (Acker and Bricker, 1992; Brantley et al., 2008; Bibi et al., 2011; Bray et al., 2015). The stoichiometry of mineral dissolution is controlled by acid type, pH, and ionic strength, as well





as the presence or absence of organic ligands (Brantley et al., 2008; Bray et al., 2015). Therefore, ratios of dissolved

concentrations of major elements vary depending on their dissolution processes from the minerals (Acker and Bricker, 1992; Brantley et al., 2008; Bibi et al., 2011; Bray et al., 2015). However, studies on Fe dissolution from aerosols have focused exclusively on Fe and have paid little attention to other dissolved metals in aerosol particles. In general, although the stoichiometry of mineral dissolution is discussed based on the ratio of the dissolution rate of major elements versus that of Si (Acker and Bricker, 1992; Brantley et al., 2008; Bibi et al., 2011; Bray et al., 2015), data on total and dissolved Si

concentrations in aerosol particles are limited relative to those on Fe and Al concentrations (Jickells et al., 2016). Discussing the dissolution processes of mineral dust by using Mg concentrations is difficult because Mg in marine aerosol particles is usually derived from sea spray aerosol (SSA). Therefore, this study focused on the molar ratio of dissolved Fe to Al ([d-Fe]/[d-Al]) in aerosol particles as the indicator of the sources of d-Fe in aerosol particles because (i) previous studies have frequently determined d-Fe and d-Al concentrations in aerosol particles and (ii) mineral dust is the dominant source of Fe and Al.

Iron-bearing particles derived from mineral dust and anthropogenic Fe have different chemical compositions and mineralogy. Mineral dust is mainly composed of crystalline aluminosilicates (Jeong and Achterberg, 2014; Jeong et al., 2014; Jeong, 2020). Fly ash emitted from anthropogenic high-temperature combustion processes can be categorized into two groups: non-magnetic and magnetic particles. Non-magnetic particles in fly ash consist of aluminosilicate glass (Furuya et al., 1987; Rivera et al., 2015). The dominant Fe species in this fraction is poorly ordered polymerized hydroxyl Fe(III) (Rivera et al.,

2015). The magnetic fraction is composed of crystalline Fe oxides (e.g., hematite and magnetite) formed through the condensation of evaporated Fe during gas cooling (Kukier et al., 2003; Fomenko et al., 2019; Czech, 2022). Both particle types have been found in ambient aerosol particles (Li et al., 2017; Zhu, Y. et al., 2020, 2022) and have higher $Fe_{sol}\%$ than mineral dust (Kurisu et al., 2016a, 2019). Therefore, the differences in mineralogy and chemical composition between mineral dust, non-magnetic, and magnetic particles likely affect the [d-Fe]/[d-Al] ratio in aerosol particles. For example, the [d-Fe]/[d-Al]

ratio of clay minerals ranges from 0.100 to 1.00 (Kodama and Schnitzer, 1973; Lowson et al., 2005; Bibi et al., 2011; Bray et al., 2015), whereas that of coal and municipal solid waste incinerators (MSWIs) is less than 0.1 (Seidel and Zimmels, 1998; Kim et al., 2003; Huang et al., 2007). In addition, the [d-Fe]/[d-Al] ratio of the aggregates of Fe oxide nanoparticles derived from anthropogenic emissions is expected to be higher than that of mineral dust and fly ash because Fe oxide nanoparticles have minimal amounts of coexisting elements (Kukier et al., 2003; Fomenko et al., 2019). The fractions of mineral dust and

anthropogenic Fe in d-Fe in aerosol particles can be estimated on the basis of the [d-Fe]/[d-Al] ratio if the [d-Fe]/[d-Al] ratio of mineral dust and anthropogenic Fe is generalized.

In 2013, the annual observation of Fe, Al, and other trace metals and their fractional solubilities in size-fractionated aerosol particles was conducted in Higashi-Hiroshima, Hiroshima, Japan. Our previous study identified that fine aerosol particles collected at the sampling site contained anthropogenic Fe with a negative $\delta^{56}Fe$ (Kurisu et al. 2016a). Therefore, the

sampling site is useful for evaluating the availability of the [d-Fe]/[d-Al] ratio as an indicator of the fractions of mineral dust and anthropogenic Fe in d-Fe in aerosol particles. In addition, the ground-based long-term observations of $Fe_{sol}\%$ are important for complementing the observed data on $Fe_{sol}\%$ in marine aerosols because the representativeness of the $Fe_{sol}\%$ data obtained



by ship-board observation is often problematic due to the difficulties in the long-term observation of marine aerosol particles at fixed stations (Mahowald et al., 2018). The ground-based observation of Fe$_{sol}$% in aerosol particles is a strategy for obtaining

long-term data on Fe$_{sol}$% because mineral dust and anthropogenic Fe (excluding ship emissions) are emitted in continental regions. In addition, Fe-bearing particles internally mixed with sulfate, nitrate, and organics are frequently found in the continental atmosphere, indicating that the chemical aging of Fe-bearing particles begins during transport in continental regions. In the chemical aging of Fe-bearing particles during transport from East Asia to the Pacific Ocean, atmospheric processing after passing over Japan has a small effect on the Fe$_{sol}$% of marine aerosol particles (Buck et al., 2013; Sakata et al., 2022).

Given that Japan is located at the rim of East Asia (or the entrance of the North Pacific Ocean), it can collect mineral dust and anthropogenic Fe aged during transport from East Asia to Japan. Therefore, it can be an important observation site for the characterization of aerosols transported to the North Pacific Ocean. This study aims to evaluate the availability of the source identification of d-Fe in aerosol particles based on [d-Fe]/[d-Al] ratios and to understand the seasonal variability of the fraction of mineral dust and anthropogenic Fe in d-Fe in aerosols.


## 2. Method

### 2.1. Aerosol sampling

Aerosol sampling was performed at Higashi-Hiroshima, Hiroshima, Japan (Fig. 1, 34.40°N, 132.71°E). Size-fractionated aerosol particles were collected by using a high-volume air sampler (MODEL 123-SL, Kimoto, Japan) with a sierra-type

cascade impactor (TE-236, Tisch Environmental Inc., USA) installed on a roof 10 m above ground level. The cascade impactor had seven stages, and the aerodynamic diameters of the aerosol particles collected in each stage from stages 1 to 7 followed the order of >10.2, 4.2–10.2, 2.1–4.2, 1.3–2.1, 0.69–1.3, 0.39–0.69, <0.39 μm. Coarse aerosol particles were coarser than 1.3 μm (stages 1 to 4), whereas fine aerosol particles were finer than 1.3 μm (stages 5 to 7). Aerosol particles were collected onto cellulose filters (stages 1 to 6: TE-230WH, Tisch Environmental Inc., USA, and S-7: Whatman 41, 8 × 10 inches, GE

Healthcare, USA). Aerosol samplings during non-dust events were conducted every month from December 2012 to December 2013 (Table S1). The flow rate of the air sampler was fixed at 0.566 m$^3$/min. The typical sampling period was 2 weeks. However, if the sampling flow rate decreased due to the clogging of the stage-7 filter, aerosol sampling was immediately stopped. Aerosol samplings were also performed during two dust events associated with haze and Asian dust in January 31, 2013 to February 1, 2013 and March 4, 2013 to March 9, 2013, respectively. The sampling periods for the polluted and Asian

dust events were 1 and 5 days, respectively, and were decided on the basis of a chemical weather forecasting system, which predicted the mass concentrations of Asian dust and sulfate aerosols (Uno et al., 2004). Serious haze events occurred in East Asia at the end of January when aerosol mass concentrations exceeded 600 μg/m$^3$ (Tian et al., 2014; Wang et al., 2014). Therefore, the haze sample was influenced by the serious air pollution event in East Asia. The backward trajectories at the altitude of 100 m were calculated by using the Hybrid Single-Particle Lagrangian-Integrated Trajectory model (Stein et al.,

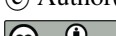



2015). Global Data Assimilation System (GDAS 0.5 degree archive) was used to collect the meteorology data. The total run time was 72 h, and the backward trajectory was calculated every 6 h. Backward trajectories for non-dust and dust events are shown in Figs. S1 and S2, respectively.

## 2.2. Major ion concentrations

Aerosol particles on approximately one-fourth of the filter strips were used for the extraction of major ions ($Na^+$, $NH_4^+$, $K^+$, $Mg^{2+}$, $Ca^{2+}$, $Cl^-$, $NO_3^-$, and $SO_4^{2-}$) by using 5 mL of ultrapure water (MQ, Merck Millipore, USA) in polypropylene vials with ultrasonication for 30 min. Suspended particles in the extract were removed by using a hydrophilic polytetrafluoroethylene (PTFE) filter ($\phi$: 0.20 μm, DISMIC, ADVANTEC, Tokyo Roshi Kaisha, Ltd., Japan). All materials used for the extraction of major ions were rinsed with MQ water three times before use.

Major ion concentrations were determined through ion chromatography (ICS–1100, Dionex Japan, Japan). The guard columns for cations and anions were Dionex Ion Pack CG12A and AG22, respectively. The separation columns for cations and anions were Dionex CS12A and AS22, respectively. The eluents for cations and anions were methanesulfonic acids and the mixed solution of 4.5 mmol/L $Na_2CO_3$ and 1.4 mmol/L $NaHCO_3$, respectively. The detail procedure of ion chromatography was described by Sakata et al. (2014).

Non-sea-salt (nss) $K^+$, $Mg^{2+}$, $Ca^{2+}$, and $SO_4^{2-}$ concentrations were estimated as follows:

$$[nss\text{-}X]_{aerosol} = [Total\text{-}X]_{aerosol} - [Na]_{aerosol} \times ([X]/[Na^+])_{seawater}, \text{ (Eq. 1)}$$

where X is the molar concentration of either $K^+$, $Mg^{2+}$, $Ca^{2+}$, or $SO_4^{2-}$. All the quantitative data of the major ions in size-fractionated aerosol particles are shown in Table S2.

## 2.3. Estimation of aerosol pH

The aerosol pH of fine aerosol particles was estimated by using the thermodynamic model E-AIM Model IV (Clegg et al., 1998; Friese and Ebel, 2010). The calculation parameters for the model were temperature, relative humidity, $[H^+]$, $[NH_4^+]$, $[Na^+]$, $[Cl^-]$, $[NO_3^-]$, and $[SO_4^{2-}]$. Ammonia gas concentration was not measured during the sampling campaign. Therefore, aerosol pH was calculated through a reverse mode that tended to overestimate aerosol acidity (Song and Osada, 2020). Previous

studies have reported that nonvolatile cations (e.g., $Mg^{2+}$ and $Ca^{2+}$) affect the calculation results of aerosol pH (Guo et al., 2018; Pye et al., 2020). $[Na^+]$ was used as a representative of nonvolatile cations (= $[Na^+] + [K^+] + 2 \times [Mg^{2+}] + 2 \times [Ca^{2+}]$) to estimate aerosol pH because the E-AIM model IV cannot incorporate the concentrations of nonvolatile cations other than $Na^+$ (Tao and Murphy, 2019a). $[H^+]$ concentration in aerosol particles was estimated on the basis of the charge balance. Several samples had negative $[H^+]$ concentrations and were thus excluded from the estimation of aerosol pH. Aerosol pH was

calculated by using the following equation:

$$\text{Aerosol pH} = -\log_{10}(m_{H+} \times \gamma_{H+}), \text{ (Eq. 2)}$$

where $m_{H+}$ and $\gamma_{H+}$ are the molar fraction and activity coefficient of $H^+$, respectively.



## 2.4. Trace metal concentrations

The target trace metals in this study were Al, Ti, V, Mn, Fe, Co, Ni, Cu, Zn, Cd, Sb, and Pb. Approximately one-fourth of the filter strips was used for acid digestion to determine total trace metal concentrations. Each filter piece was digested with a mixed acid solution of 15.2 mol/L $HNO_3$, 9.3 mol/L HCl, and 20 mol/L HF in 7 mL of perfluoroalkoxyalkane vial heated at 120 °C for 12 h (all mineral acids were procured from TAMAPURE AA-100, Tama Chemicals Co., Ltd., Japan). Subsequently, the mixed acid was evaporated to dryness. The dried residue was dissolved in 5 mL of 0.15 mol/L $HNO_3$ heated at 120 °C for

3 h. The solution was filtered through a hydrophilic PTFE filter ($\phi$: 0.20 µm, Dismic®, 25HP020AN, Advantec, Japan) to prepare the analytical solution. Soluble metals in aerosol particles were extracted by using 5 mL of MQ water with ultrasonication for 30 min. Then, the extracted solution was filtrated with a hydrophilic syringe PTFE filter. The filtrated solution was acidified to prepare a 0.15 mol/L $HNO_3$ solution by adding 15.3 mol/L $HNO_3$. The filtrated solutions for total and dissolved metal concentration analyses were diluted with 0.15 mol/L $HNO_3$ by factors of 10–1000.

The total and dissolved metal concentrations of the sample solution were determined via inductively coupled plasma mass spectrometry (ICP–MS, Agilent 7700, Agilent, Japan). The sample introduction system consisted of a borosilicate nebulizer (MicroMist, Agilent, Japan) and a quartz Scott double-pass spray chamber. The elements in the sample solution were ionized by an argon plasma (RF power: 1500 W, Ar flow for plasma: 1.0 L/min). The ions were introduced into the detection system by passing through Ni sampling and skimmer corns. The ion beam was focused by using x-type ion lenses. The He collision

mode was employed to reduce interference from oxides (e.g., $^{40}Ar^{16}O^+$ for $^{56}Fe$). Helium gas was injected into a dynamic reaction cell at a rate of 3.6 mL/min. Then, mass selection was performed by using a quadrupole system. Target metal concentrations were measured in pulse (ion-counting) mode. The sensitivity drift during the measurements was corrected by using the internal standard of 1 ng/g of In. All quantitative data on total and dissolved Fe and Al concentrations and their fractional solubilities are shown in Table S3. The total and dissolved metal concentrations of other target elements are shown

in Tables S4 and S5, respectively.

The total and dissolved metal concentrations in TSP were calculated by the summation of the target metal concentrations in all sampling stages. Crustal and non-crustal Fe concentrations and the EF of target metals normalized by the average continental crust were calculated by using the following equations:

$$Crustal\ Fe = Al_{crust} \times (Fe/Al)_{aerosol}, (Eq.\ 3)$$

$$Non\text{-}crustal\ Fe = Total\ Fe - crustal\ Fe, (Eq.\ 4)$$

$$EF = (M/Al)_{aerosol}/(M/Al)_{crust}, (Eq.\ 5)$$

where M is the target metal. Their concentrations in the average continental crust were acquired by referring to Taylor (1964). The fractional solubility of target metals in aerosol samples was calculated by using the following equation:

$$Fractional\ solubility\ (\%) = (Dissolved\ M/total\ M) \times 100. (Eq.\ 6)$$




## 3.    Results and Discussion

### 3.1. Major ion concentrations

#### 3.1.1.    Cations

Sodium ion, $Mg^{2+}$, and $Ca^{2+}$ were mainly distributed in coarse aerosol particles, accounting for 88.3 %, 84.8 %, and 79.3
% of the ions in TSP, respectively (Figs. 2a–2c). Sodium ion in aerosol particles was mainly associated with sea spray aerosol
(SSA). Magnesium ion was mainly derived from SSA considering that nss-$Mg^{2+}$ accounted for $26.2 \pm 22.4$ % of the total $Mg^{2+}$.
By contrast, almost all $Ca^{2+}$ ($90.8 \pm 9.45$ %) was present in the form of nss-$Ca^{2+}$. Calcium ion concentration was higher in
spring (March to May) than in other seasons and during Asian dust events (Fig. 2b). A large amount of Asian dust is transported
from the Gobi or Taklamakan Deserts in spring (Uematsu et al., 1983; Sullivan et al., 2007a). Therefore, the high $Ca^{2+}$
concentration in spring was attributed to Asian dust. Potassium ion and $NH_4^+$ were mainly distributed in fine aerosol particles,
which accounted for $68.2 \pm 9.69$ % and $83.0 \pm 3.49$ % of the ions in TSP, respectively (Figs. 2d and 2e). More than 90 % of
$K^+$ in fine aerosol particles (annual average: $94.5 \pm 14.8$ %) was present in the form of nss-$K^+$. The nss-$K^+$ in fine aerosol
particles is mainly derived from either biomass burning or coal combustion (Echalar et al., 1995; Simoneit et al., 2002; Yu et
al., 2018). The discussion on the size and seasonal variation of $NH_4^+$ with $NO_3^-$ and $SO_4^{2-}$ is provided in the next section.

#### 3.1.2.    Anions

Chloride ion dominated in coarse aerosol particles, which contributed $79.5 \pm 14.1$ % of $Cl^-$ in TSP (Fig. 2f). SSA is the
dominant source of $Cl^-$ in aerosol particles. However, the $Cl^-/Na^+$ mass ratio of aerosol particles was not identical to that of
seawater (Fig. S3a). Chloride ion concentration in coarse aerosol particles was depleted relative to the expected $Cl^-$
concentration in non-aged SSA (= $Na^+_{aerosol} \times [Cl^-/Na^+]_{seawater}$), and the depletion ratio of $Cl^-$ in coarse aerosol particles to $Cl^-$
in non-aged SSA was $34.7\% \pm 28.2\%$. $Cl^-$ depletion was caused by the chemical reaction of NaCl with $HNO_3$ and $H_2SO_4$ as
follows (Finlayson-Pitts, 2003):

$$NaCl + HNO_3 \rightarrow NaNO_3 + HCl(g), \text{ (R1)}$$

$$2NaCl + H_2SO_4 \rightarrow Na_2SO_4 + 2HCl \text{ (g). (R2)}$$

The frequent enrichment of $Cl^-$ in fine aerosol particles relative to that in non-aged SSA (Fig. S3a) and in contrast to that in
coarse aerosol particles indicated that emission sources other than SSA contributed to $Cl^-$ in fine aerosol particles. $Cl^-$
enrichment was observed in aerosol samples collected in winter and spring when air masses mainly originated from East Asia
(Fig. S1 and S3a). Previous studies have reported that anthropogenic emissions (e.g., coal combustion, industrial processes,
and MSWI) and biomass burning are the dominant sources of HCl and $Cl^-$ in fine aerosol particles in East Asia (Fu et al.,
2018; Liu et al., 2018). Indeed, the correlation of excess $Cl^-$ concentration (= $-1 \times Cl^-$ loss) with nss-$K^+$ is a tracer of biomass
burning and coal combustion (r: 0.570). In addition, pre-existing particles, including $CaCO_3$ in mineral dust, act as the sink of
Cl species (Sullivan et al., 2007b; Tobo et al., 2010). Therefore, the enrichment of $Cl^-$ in fine aerosol particles was caused by
the uptake of anthropogenic $Cl^-$ by pre-existing particles.



Sulfate ions and $NH_4^+$ were mainly distributed in fine aerosol particles (Figs. 2e and 2g) and accounted for $75.8 \pm 11.1$ %
and $88.8 \pm 7.68$ % of the total anions and cations in fine aerosol particles, respectively. The average fraction of nss-$SO_4^{2-}$ to

total $SO_4^{2-}$ (nss-$SO_4^{2-}$/total $SO_4^{2-}$) in coarse and fine aerosol particles were $70.1 \pm 23.1$ % and $99.3 \pm 1.44$ %, respectively.

Thus, nss-$SO_4^{2-}$ was dominant in coarse and fine aerosol particles. Ammonium ion concentration had a good correlation with

but was higher than nss-$SO_4^{2-}$ concentration (Fig. S3b). This result indicated that ammonium salts other than $(NH_4)_2SO_4$ and

$NH_4HSO_4$ were present in fine aerosol particles. Ammonium ion concentration was found to have an excellent correlation with

$2 \times$ [nss-$SO_4^{2-}$] + [$NO_3^-$] in fine aerosol particles (Fig. S3c). The slope of the regression line was 0.965, indicating that

$(NH_4)_2SO_4$ and $NH_4NO_3$ were the dominant major ion components in fine aerosol particles.

Nitrate ion had concentration peaks not only in fine aerosol particles but also in coarse aerosol particles (Fig. 2h). The

average fractions of $NO_3^-$ in coarse and fine aerosol particles in TSP were $61.3 \pm 12.3$ % and $36.9 \pm 10.5$ %, respectively. As

previously mentioned, $NO_3^-$ in coarse aerosol particles was derived from $Cl^-$ depletion as described in R1. Assuming that

$NO_3^-$ caused the depletion of all Cl from SSA in coarse aerosol particles, SSA-associated $NO_3^-$ accounted for only $35.1 \pm 25.1$

% of $NO_3^-$ in coarse aerosol particles. Therefore, $NO_3^-$ ions were mainly present in coarse aerosol particles other than SSA

(non-SSA-$NO_3^-$). Previous studies have reported that mineral dust is the dominant driver of $NO_3^-$ concentration in coarse

aerosol particles (Karydis et al., 2016, Kakavas et al., 2021). In fact, the good correlation between nss-$Ca^{2+}$ and non-SSA-

$NO_3^-$ (= total $NO_3^-$ − Cl depletion) found in our coarse aerosol particles (Fig. S3d) indicated that non-SSA-$NO_3^-$ was present

in coarse aerosol particles in the form of $Ca(NO_3)_2$. However, our previous study identified gypsum ($CaSO_4 \cdot 2H_2O$) rather than

$Ca(NO_3)_2$ as the dominant secondary Ca species in coarse aerosol particles collected in January, November, and the Asian dust

event (Miyamoto et al., 2020). Recent studies have demonstrated that hygroscopic $Ca(NO_3)_2$ on the surfaces of mineral dust

reacted with $(NH_4)_2SO_4$, resulting in the formation of $NH_4NO_3$ and $CaSO_4 \cdot 2H_2O$ (Wu et al., 2019, 2020). Thus, $NO_3^-$ taken

up in the reaction with $CaCO_3$ is considered to exist in the form of $NH_4NO_3$ instead of $Ca(NO_3)_2$.


### 3.2. Total Fe and Al concentrations

The total Fe and Al concentrations in TSP collected during the non-dust event were 256–1561 and 170–1716 ng m$^{-3}$,

respectively (Figs. 3a and 3b). Coarse aerosol particles accounted for $69.4 \pm 7.19$ % and $72.9 \pm 7.55$ % of the total Fe and Al

in TSP, respectively. The total Fe and Al concentrations were higher in spring (March to May) than in other seasons due to the

strong influence of Asian dust in spring (Figs. 3a and 3b). The total Fe and Al concentrations in the samples collected during

the Asian dust event were considerably higher than those in the samples collected during the non-dust event (Figs. 3a and 3b).

The EF of Fe in TSP samples collected in summer, when air masses mainly originated from Japan, was higher than of Fe in

TSP samples collected in other seasons (Figs. S1 and S4a). The total Al concentrations sharply decreased from May to June,

whereas the total Fe concentrations did not decrease as significantly as the total Al (Figs. 3a and 3b). This result indicated that

aerosol particles emitted in Japan contained non-crustal Fe. In addition, the total Fe concentration in the haze event was almost

same as that in the Asian dust event. However, the total Al concentration in the haze event was lower than that in the Asian





dust event. This result demonstrated that during haze events, total Fe concentrations were significantly higher than total Al concentrations due to the influence of non-crustal Fe.

The average EFs of Fe in coarse and fine aerosol particles were 1.99 ± 0.892 and 2.49 ± 1.09, respectively (Fig. 3c).
Therefore, non-crustal Fe may be present in coarse and fine aerosol particles. Non-crustal Fe concentration were correlated with Cu, Zn, Sb, and Pb in road dust (Figs. 4a–d). The elements in coarse aerosol particles are known as the tracer elements of fragments vehicle-related materials in road dust (brake dust: Cu and Sb, tire wear: Zn, and road paint: Pb) (Adachi and Tainosho, 2004; Wåhlin et al., 2006; Iijima et al., 2007; Gietl et al., 2010; Sakata et al., 2014; Harrison et al., 2021). A correlation coefficient of non-crustal Fe with Cu as a tracer of brake pad was the highest among these tracer elements. It is
known that Fe is the most dominant metals in brake rings (up to 50 wt%) and its concentration is about an order of magnitude higher than Al in the brake pad. Therefore, the brake pad can increase EF of Fe in aerosol particles. In fact, EF of Fe in road dust (<40 μm) collected from tunnels and roadsides around the sampling site were 3.41 and 6.04, respectively. Therefore, the resuspensions of brake pad fragments in road dust could be considered as non-crustal Fe in coarse aerosol particles. This result is consistent with the finding of a previous model study because road dust is the dominant source of Cu and non-crustal Fe in
$PM_{10}$ in the Chugoku–Shikoku area (the locality around the sampling site) (Kajino et al., 2020).

In fine aerosol particles, non-crustal Fe concentrations were correlated with Zn, Sb, and Pb (Fig. 5a–c), which were emitted by coal combustion, MSWI, and steel/iron industrial processes (Pacyna and Pacyna, 2001; Sakata et al., 2000; Sakata et al., 2014; Kajino et al., 2020). These high-temperature combustion processes emit nanoparticles of hematite and magnetite with a small number of coexisting elements, which can increase the EF of Fe in fine aerosol particles. Indeed, in this study,
the negative $\delta^{56}Fe$ associated with hematite derived from high-temperature combustion processes was detected in aerosol samples collected at the same sampling (Kurisu et al., 2016a). Therefore, the nanoparticles of hematite and magnetite are likely the dominant source of non-crustal Fe in fine aerosol particles. In addition, in fine aerosol particles, the non-crustal Fe concentration was weakly correlated with V (Fig. 5d), which is a tracer of heavy-oil combustion (Nriagu and Pacyna, 1988). A previous model study predicted that V around the sampling site was derived from vessel emissions in the Seto Inland Sea
located at the southern part of the sampling site (Figure 1, Kajino et al., 2020). Our results also showed that V concentrations in fine aerosol particles were higher from July to September, during which air masses pass through the Seto Inland Sea (Figs. 1 and S5a), than in other periods. At this time, the concentrations of Zn, Sb, and Pb did not increase (Fig. S5b–d). Therefore, in addition to the high-temperature combustion processes mentioned above, heavy-oil combustion processes around the Seto Inland Sea likely were the emission sources of non-crustal Fe in fine aerosol particles collected in summer.


### 3.3. Size-distributions of d-Fe and d-Al concentration and their fractional solubility

Dissolved Fe and Al concentrations in TSP collected during the non-dust event ranged from 11.4 mg m⁻³ to 65.0 ng m⁻³ and from 8.30 ng m⁻³ to 40.6 ng m⁻³, respectively (Figs. 6a and 6d). The $Fe_{sol}$% and $Al_{sol}$% of TSP ranged from 2.00 to 7.73 % and 1.46 to 7.39 %, respectively (Figs. S4b and S4c). Although approximately 70 % of the total Fe and Al in TSP were present
in coarse aerosol particles, high concentrations of d-Fe and d-Al were found in fine aerosol particles (Figs. 6a and 6d). The d-





Fe and d-Al of fine aerosol particles were $72.0 \pm 8.87$ % and $53.1 \pm 9.90$ %, respectively. Thus, d-Fe and d-Al concentrations yielded different size distributions from total Fe and Al concentrations. The reason for the enrichment of d-Fe and d-Al in fine aerosol particles was higher $Fe_{sol}$% and $Al_{sol}$% in fine aerosol particles than coarse aerosol particles. The average $Fe_{sol}$% in fine aerosol particles ($11.4 \pm 6.97$%) was about five times higher than that of coarse aerosol particles ($2.19 \pm 2.27$ %, Figs. 6b and

6c). In the case of Al, fine aerosol particles ($8.82 \pm 6.48$ %) yielded twice as much $Al_{sol}$% as coarse aerosol particles ($3.25 \pm 3.41$ %, Figs. 6e and 6f). Enrichment of d-Fe and d-Al in size-fractionated aerosol samples has been reported not only in the urban atmosphere (Fang et al., 2017; Hsieh et al., 2022) but also in the marine atmosphere (McDaniel et al., 2019; Baker et al., 2020; Sakata et al., 2022). These previous studies have reported that the higher d-Fe and d-Al in coarse aerosol particles were caused by either or both (i) contamination of anthropogenic aerosol with high $Fe_{sol}$% and $Al_{sol}$% and (ii) solubilization of

Fe and Al in fine aerosol particles by atmospheric processes. It should be noted that $Fe_{sol}$% and $Al_{sol}$% in coarse aerosol particles were higher than in desert dust soil (typical $Fe_{sol}$% and $Al_{sol}$%: 0.1%). Therefore, the $Fe_{sol}$% and $Al_{sol}$% in coarse aerosol particles were increased by anthropogenic aerosol and atmospheric processes, but the effect was not as pronounced as for fine particles.

**3.4. Possible factor controlling $Fe_{sol}$% and $Al_{sol}$%**

**3.4.1.    Coarse aerosol particles**

Non-crustal Fe in coarse aerosol particles was derived from brake bad and tire ware debris in road dust. If road dust is the dominant source of d-Fe in coarse aerosol particles, d-Fe concentration is correlated with non-crustal Fe concentration. However, a correlation between non-crustal Fe and d-Fe concentration in coarse aerosol particles was not found (r: 0.352),

indicating that road dust was not the source of d-Fe in coarse aerosol particles. This result is consistent with previous studies because $Fe_{sol}$% of debris of brake pads and tire wear is lower than 0.01% (Shupert et al., 2013; Halle et al., 2021).

The $Fe_{sol}$% and $Al_{sol}$% of coarse aerosol particles increased with the decrease in aerosol diameter (Fig. 6b, 6c, 6e, and 6f). Specific surface area is one of the factors controlling $Fe_{sol}$% and $Al_{sol}$% in aerosol particles (Baker and Jickells, 2006, 2017; McDaniel et al., 2019), and the chemical reactivity of aerosol particles is increased with increasing specific surface area

(= decreasing aerosol diameter). Previous studies have reported that the chemical aging of coarse aerosol particles by acidic species, including $HNO_3$ and $H_2SO_4$, were solubilized Fe in aerosol particles (Takahashi et al., 2011; Zhu et al., 2022). The $Fe_{sol}$% and $Al_{sol}$% in coarse aerosol particles correlated with nss-$SO_4^{2-}$/total Al and nss-$SO_4^{2-}$/total Fe (Fig. 7a). This result is consistent with previous findings because aluminosilicates in aerosol particles react preferentially with $H_2SO_4$ (Sullivan et al., 2007a; Fitzgerald et al., 2015). Furthermore, our previous studies showed that a relative abundance of ferrihydrite formed by

hydrolysis of Fe in coarse aerosol particles was increased with decreasing diameter (Takahashi et al., 2011; Sakata et al., 2012). These results indicated that atmospheric processes of coarse aerosol particles by $H_2SO_4$ promote the formation of d-Fe via proton-promoted dissolution or hydrolysis of Fe. By contrast, no good correlation was found between $Fe_{sol}$% and $Al_{sol}$% and between $NO_3^-$/total Fe and $NO_3^-$/total Al because $HNO_3$ reacts mainly with Ca-rich particles (e.g., $CaCO_3$) in mineral dust, as mentioned above (Fig. 7b, Karydis et al., 2016; Kakavas et al., 2021).





### 3.4.2.    Fine aerosol particles


Unlike d-Fe in coarse aerosol particles, d-Fe concentrations in fine aerosol particles correlated with non-crustal Fe concentrations (Fig.8a). Good correlations of d-Fe in fine aerosol particles with concentrations of V, Zn, Sb, and Pb as tracer elements for high-temperature combustions were also found (Fig. 8b-8e). Among these tracer elements, the correlation coefficient between d-Fe and Zn was the largest (Fig. 8c). Since Zn in fine aerosol particles at the sampling sites originated

from various high-temperature combustions, including MSWI, steel/iron industries, and coal power plant (Kajino et al., 2020). Therefore, high-temperature combustion is one of the dominant sources of d-Fe in fine aerosol particles, though it is not easy to identify specific sources of d-Fe around the sampling site. This result is consistent with our previous studies because negative $\delta^{56}$Fe associated with high-temperature combustions were detected in fine aerosol particles (Kurisu et al., 2016a).

It is known that fine aerosol particles usually yield lower aerosol pH compared to coarse aerosol particles (Pye et al.,

2020). Therefore, as well as coarse aerosol particles, atmospheric processes also contribute to solubilizing Fe in fine aerosol particles. The $Fe_{sol}$% and $Al_{sol}$% of fine aerosol particles were weakly correlated with nss-$SO_4^{2-}$/Fe and nss-$SO_4^{2-}$/Al, respectively, but not with $NO_3^-$/Fe and $NO_3^-$/Al (Figs. 7c and 7d). In addition, $Fe_{sol}$% and $Al_{sol}$% tended to increase with the decrease in aerosol pH (Fig.7e). These results implied that the $Fe_{sol}$% and $Al_{sol}$% of fine aerosol particles increased via aerosol acidification, which occurred through the chemical reaction of Fe- and Al-bearing particles with $SO_2$ and $H_2SO_4$. The

correlation factor between $Al_{sol}$% and nss-$SO_4^{2-}$/Al was higher than that between $Fe_{sol}$% and nss-$SO_4^{2-}$/Fe (Figs. 7c and 7d). In addition, the correlation factor of the $Fe_{sol}$% and nss-$SO_4^{2-}$/Fe of fine aerosol particles was lower than that of coarse aerosol particles (Figs. 7a and 7c). Given that Al in fine aerosol particles was derived from aluminosilicates, $Al_{sol}$% was mainly controlled by the acidification of mineral dust. These results are reasonable because previous studies have reported that in fine aerosol particles, mineral dust was covered with sulfates (Sullivan et al., 2007a; Fitzgerald et al., 2015; Li et al., 2017, Zhu, Y.

et al., 2020, 2022). By contrast, the low correlation between the $Fe_{sol}$% and nss-$SO_4^{2-}$/Fe of fine aerosol particles implied that the acidification of mineral dust was not the sole factor controlling $Fe_{sol}$%.

### 3.5. The [d-Fe]/[d-Al] ratio in coarse aerosol particles and mineral dust

Thus, non-crustal Fe contributed to the source of d-Fe in fine aerosol particles but not in coarse aerosol particles.

Correlation analysis is a simple method to evaluate emission sources of d-Fe in aerosol particles, but quantitative estimation of the relative abundance of non-crustal Fe to d-Fe in aerosol particles is difficult. Therefore, it is preferable to have a quantitative indicator to assess the relative abundance of non-crustal Fe to d-Fe. This study challenged the evaluation of the relative abundance of non-crustal Fe to d-Fe using the [d-Fe]/[d-Al] ratio. One of the reasons is that that the [d-Fe]/[d-Al] ratio in mineral dust differs from non-crustal sources discussed below (Fig. 9a). In fact, the [d-Fe]/[d-Al] ratio in fine aerosol

particles (average: 1.15 ± 0.803) was different from that in coarse aerosol particles (Fig. 9b, average: 0.408 ± 0.168).

Evaluating the relative abundance of non-crustal Fe to d-Fe in aerosol particles, it is necessary to generalize the [d-Fe]/[d-Al] ratios of crustal and non-crustal Fe. Therefore, we performed data compiles of the [d-Fe]/[d-Al] ratio in mineral dust and non-crustal Fe sources. In addition, the [d-Fe]/[d-Al] ratio may vary depending on the extraction method as well as the chemical





weathering process in the atmosphere (Sholkovitz et al., 2012; Clough et al., 2019). Therefore, the data of [d-Fe]/[d-Al] were

compiled considering not only the differences in the emission sources of Fe in aerosols but also the differences in the dissolution processes (i.e., proton-promoted vs. ligand-promoted dissolutions). The results are summarized in Fig. 9a.

In the present study, we focused on biotite, illite, and Fe-rich chlorite as the representative mineral species of aluminosilicates in aerosol particles because these aluminosilicates were detected in aerosol particles collected in East Asia (Takahashi et al., 2011; Kurisu et al., 2016a; Sakata et al., 2022). When Fe and Al in these aluminosilicates were leached by

proton-promoted dissolution (pH 1.0–7.0) under oxic conditions, the [d-Fe]/[d-Al] ratios of biotite, illite, and Fe-rich chlorite was 0.427–0.930 (average: $0.776 \pm 0.152$), 0.156–0.689 (average: $0.404 \pm 0.189$), and 0.142–1.03 (average: $0.699 \pm 0.303$), respectively (Fig. 9a; Kodama and Schnitzer, 1973; Lowson et al., 2005; Bibi et al., 2015; Bray et al., 2015). The [d-Fe]/[d-Al] ratio tended to decrease with the increase in pH due to either or both the preferential retention of Fe in the mineral phase and precipitation of secondary ferrihydrite under near-neutral conditions (Fig. S6, Kodama and Schnitzer, 1973; Desboeufs et

al., 2001; Lowson et al., 2005; Bray et al., 2015). In addition, the average [d-Fe]/[d-Al] ratios of Asian dust, Saharan dust, and Arizona test dust obtained by proton-promoted dissolution were $0.238 \pm 0.201$, $0.163 \pm 0.157$, and $0.230 \pm 0.00926$, respectively (Fig. 8a, Desboeufs et al., 2001; Duvall et al., 2008; Shi et al., 2011). The [d-Fe]/[d-Al] ratio is also decreased with increasing pH (Fig. S6, Desboeufs, et al., 2001). Thus, the [d-Fe]/[d-Al] ratio obtained from the proton-promoted dissolution of the mineral dust was between 0.1 and 1.0.

Organic ligands promote the dissolution of Fe from the mineral phase through the direct complexation of organic ligands with Fe at the mineral surface and the reduction in the saturation index of inorganic Fe due to the formation of organic complexes of Fe in solution (Chen and Grassian, 2013; Paris and Desboefus, 2013; Wang et al., 2017). The [d-Fe]/[d-Al] ratios of biotite and Fe-rich chlorite associated with ligand-promoted dissolution were 0.795–3.83 (average: $1.20 \pm 0.660$) and 1.19–1.37 (average: $1.31 \pm 0.0629$), respectively (Kodama and Schnitzer, 1973; Bray et al., 2015). A previous study found that the

[d-Fe]/[d-Al] ratios of coarse aerosol particles extracted by 20 mmol/L of oxalate at pH 4.7 ($1.31 \pm 0.418$) were higher than that of MQ extraction ($0.354 \pm 0.714$) (Coarse aerosol (Proton) and Coarse aerosol (Lignad) in Fig. 9a, Kurisu et al., 2019). Thus, the [d-Fe]/[d-Al] ratio associated with ligand-promoted dissolution was higher than that associated with proton-promoted dissolution (Fig. 8a). The higher stability constant of $Fe^{3+}$ than that of $Al^{3+}$ with organic ligands (e.g., $AlC_2O_4^+$: 7.73, $FeC_2O_4^+$: 9.15) is a reason for the preferential dissolution of Fe over Al by organic ligands, increasing the [d-Fe]/[d-Al] ratio.

The [d-Fe]/[d-Al] ratio of coarse aerosol particles (0.121–0.927) was within a range of the [d-Fe]/[d-Al] of mineral dust that underwent proton-promoted dissolutions (Fig. 9b). In addition, the [d-Fe]/[d-Al] ratio of coarse aerosol particles increased with the increase in $Fe_{sol}$% (Fig. 9b, r: 0.552). Coarse aerosol particles with high $Fe_{sol}$% yielded high $nss-SO_4^{2-}$/total Fe (Fig. 7a), indicating that coarse aerosol particles with high $Fe_{sol}$% underwent acidic conditions. This result is consistent with a relationship of the [d-Fe]/[d-Al] ratio with dissolution pH. Therefore, the proton-promoted dissolution of mineral dust is the

dominant source of d-Fe in coarse aerosol particles. On the other hand, no coarse aerosol particles had [d-Fe]/[d-Al] ratios higher than 1.0 (Fig. 9b). Therefore, the effect of organic ligands on the solubilization of Fe and Al in coarse aerosol particles was not significant. In addition, non-crustal Fe derived from road dust was not factors controlling the [d-Fe]/[d-Al] ratio in





coarse aerosol particles due to no correlation between the [d-Fe]/[d-Al] ratio and EF of Fe. This result is reasonable because the brake pads and tire wear debris are less than 0.01% (Shupert et al., 2013; Halle et al., 2021).


### 3.6. The [d-Fe]/[d-Al] ratio in fine aerosol particles and their emissions

The average [d-Fe]/[d-Al] ratio of fine aerosol particles (n = 45, average: 1.15 ± 0.803, range: 0.386–4.67) was higher than that of coarse aerosol particles and mineral dust (Figs. 9a and 9b). In particular, the [d-Fe]/[d-Al] ratios of several fine aerosol particles exceeded 1.5, which was beyond the range of the ratio of mineral dust that underwent ligand-promoted dissolutions (Fig. 9a and 9b). Considering a good correlation between the [d-Fe]/[d-Al] ratio and EF of Fe (r: 0.505), the [d-Fe]/[d-Al] ratio higher than 1.5 was attributed to non-crustal Fe (Fig. 9c). Non-crustal Fe in fine aerosol particles was derived from fly ash of high-temperature combustions, including steel/iron industrial processes, MSWI, and coal and fuel oil combustion. Fly ash contains two types of particles, namely magnetic and non-magnetic particles (Kukier et al., 2003; Fomenko et al., 2021). Since Fe and coexisted elemental concentrations and Fe species are different between magnetic and non-magnetic particles, the impact of these particles on the [d-Fe]/[d-Al] ratio and $Fe_{sol}$% in fine aerosol particles were evaluated separately.

### 3.6.1. [d-Fe]/[d-Al] ratio of non-crustal Fe: magnetic particles

Mass fraction of magnetic particles was only a few percent to total fly ash mass (Hansen et al., 1981), but up to 90 % of Fe in fly ash is present as magnetic particles (Kukier et al., 2003). Iron species of magnetic particles in fly ash are mainly composed of iron oxide, including hematite and magnetite. Iron concentration in magnetic particles was up to 60–70 %, whereas Al concentration in magnetic particles is typically up to 10 % (Kukier et al., 2003; Fomenko et al., 2021). This result indicated that Fe in magnetic particles in fly ash can be increased EF of fine aerosol particles.

Previous studies have reported that d-Fe in fine aerosol particles yielded negative Fe isotope ratio ($\delta^{56}Fe < 0$ ‰, Kurisu et al., 2016a, 2019) and a recent study show negative $\delta^{56}Fe$ in magnetic particles in $PM_{2.5}$ (Zuo et al., 2021). The magnetic particles in fine fractions (<1.0 μm) are considered to be formed by either or both fragmentation of large particles and condensation of vaporized Fe. In particular, Fe-oxides particles formed by condensation of vaporized Fe (hereafter high-temp-FeOx) is recognized as one of the important sources of d-Fe in fine aerosol particles because the high-temp-FeOx yield negative Fe isotope ratio caused by kinetic Fe isotope fractionation during vaporization process (Kurisu et al., 2016, 2019). In addition, the [d-Fe]/[d-Al] ratio in fine aerosol particles with low $\delta^{56}Fe$ (< −1.0 ‰) collected near steel plant was higher than 1.5 when extracted with proton-promoted (MQ) and ligand-promoted (20 mmol/L oxalic acid at pH 4.7) dissolutions (Kurisu et al., 2019). Thus, high-temp-FeOx can increase the [d-Fe]/[d-Al] ratio of fine aerosol particles. However, $Fe_{sol}$% of the magnetic particles in coal fly ash was less than 0.1 % in weakly acidic solutions (Kukier et al., 2003), which was much lower than $Fe_{sol}$% in fine aerosol particles (11.4 ± 6.97 %). As previously mentioned, the $Fe_{sol}$% of fine aerosol particles in fine aerosol particles increase with the increasing nss-$SO_4^{2-}$/total Fe and with decrease aerosol pH (Fig. 7c and 7e). In addition, aerosol pH with high





$Fe_{sol}\%$ (>10 %) was basically lower than 10, which was consistent with previous observations (Tao and Murphy, 2019b). Therefore, aerosol acidification is also the dominant factor controlling $Fe_{sol}\%$ in fine aerosol particles.

Ligand-promoted process may have partly enhanced Fe dissolution in mineral dust in fine aerosol particles with [d-Fe]/[d-Al] ratios of 1.0 to 1.5 (Fig. 9b). If ligand-promoted dissolution of mineral dust in fine aerosol particles is the dominant source

of d-Fe in fine aerosol particles, a good correlation between [Fe]/[Al] ratio and EF of Fe should not be observed because EF of mineral dust is about 1.0. Therefore, we conclude the high [Fe]/[Al] ratio in fine aerosol particles attributed to the presence of high-temp-FeOx rather than ligand-promoted dissolution of mineral dust.

### 3.6.2.    The [d-Fe]/[d-Al] ratio of non-crustal Fe: non-magnetic fraction of fly ash

Non-magnetic particles are mainly composed of amorphous aluminosilicate glasses. The amorphous aluminosilicates are considered to be formed by the melting of crystalline-aluminosilicate or chemical reactions of melted $Al_2O_3$ and $SiO_2$ (Zhang et al., 2007). The average [d-Fe]/[d-Al] ratio of coal and MSWI fly ash under acidic and circumneutral conditions was 0.104 ± 0.0751 (Fig. 9a, Seidel and Zimmels, 1998; Praharaj et al., 2002; Kim et al., 2003; Huang et al., 2007; Chang et al., 2009; Gitari et al., 2009; Komonweeraket et al., 2015). These studies determined d-Fe and d-Al concentrations in fly ash without a

separation between magnetic and non-magnetic particles. Considering the high [d-Fe]/[d-Al] ratio in the magnetic particles, it is expected that the [d-Fe]/[d-Al] ratio of non-magnetic particles is less than 0.1. However, no fine aerosol particles had a [d-Fe]/[d-Al] ratio less than 0.1 (Fig. 9b). In addition, non-magnetic aluminosilicates of coal combustions and MSWI could not increase the EF of Fe in fine aerosol particles because the EF of Fe in these ashes was almost 1 (Sakata et al., 2022; Li et al., 2022). Therefore, non-magnetic aluminosilicates derived from coal combustion and MSWI were unlikely the dominant source

of aerosol particles with high $Fe_{sol}\%$ and [d-Fe]/[d-Al] ratios.

In the case of heavy-oil combustion, $Fe_{sol}\%$ in oil fly ash can reach 80% under circumneutral pH conditions because of the presence of water-soluble Fe(III)-sulfate (Schroth et al., 2009; Oakes et al., 2012). Despite limited data, the [d-Fe]/[d-Al] ratios of oil fly ash were 0.403, 1.07, and 7.00 at pH 5.7, 4.7, and −0.3, respectively (Akita et al., 1995; Desboeufs et al., 2001). Assuming a linear relationship between the pH of heavy oil fly ash and the [d-Fe]/[d-Al] ratio, the expected [d-Fe]/[d-Al] ratio

of heavy-oil ash was approximately 4.0 at the average aerosol pH in fine aerosol particles collected in summer (average: 2.11 ± 0.452). However, it is difficult to explain the reason for the correlation between the [d-Fe]/[d-Al] ratio and EF of Fe in fine aerosol particles because the EF of Fe in heavy-oil fly ash was almost 1.0 as well as non-magnetic aluminosilicates of coal and MSWI (Sakata et al., 2017). Therefore, it is considered that the contribution of fly ash emitted from heavy oil combustion is not large.


### 3.7. Fractions of crustal Fe and high-temp-FeOx in d-Fe and their $Fe_{sol}\%$
### 3.7.1.    Estimation method for the fractions

The fractions of mineral dust and non-crustal Fe in d-Fe in fine aerosol particles were estimated based on the [d-Fe]/[d-Al] ratio of fine aerosol particles. The [d-Fe]/[d-Al] ratios of crustal Fe and non-crustal Fe were set to values generalized from



the results obtained in this study. The average [d-Fe]/[d-Al] ratio of coarse aerosol particles was 0.408, which was within the
range of the ratios of aluminosilicates and loess samples in the references (Fig. 9a and 9b). In the case of non-crustal Fe, we
only focused on high-temp-FeOx because the fine aerosol particles did not exhibit the low [d-Fe]/[d-Al] ratio attributed to non-
magnetic aluminosilicates of coal combustions and MSWI ([d-Fe]/[d-Al] < 0.1, Fig. 9a). However, the [d-Fe]/[d-Al] ratio of
high-temp-FeOx has not been reported. Therefore, two [d-Fe]/[d-Al] ratios of pyrogenic-FeOx were used in the calculation:

the non-crustal [d-Fe]/[d-Al] ratio of 4.67, which was the highest [d-Fe]/[d-Al] ratio of fine aerosol particles. Another is the
average [d-Fe]/[d-Al] ratio of 2.08 of fine aerosol particles with [d-Fe]/[d-Al] ratios higher than 1.5.

In consideration of the binary mixing of d-Fe in crustal d-Fe and high-temp-FeOx (crustal-dFe and high-temp-FeOx-
dFe), the fraction of high-temp-FeOx-dFe in d-Fe were estimated by using the following equations:

$$f_{\text{crustal-dFe}} + f_{\text{high-temp-FeOx-dFe}} = 1, \text{ (Eq. 7)}$$

$$[\text{d-Fe}]/[\text{d-Al}]_{\text{aerosol}} = ([\text{d-Fe}]/[\text{d-Al}])_{\text{crust}} \times f_{\text{crustal-dFe}} + ([\text{d-Fe}]/[\text{d-Al}])_{\text{non-crust}} \times f_{\text{high-temp-FeOx-dFe}}, \text{ (Eq. 8)}$$

where $f_{\text{crustal-Fe}}$ and $f_{\text{high-temp-FeOx-dFe}}$ are the fractions of d-Fe derived from crustal Fe and high-temp-FeOx, respectively. ([d-
Fe]/[d-Al])$_{\text{aerosol}}$, ([d-Fe]/[d-Al])$_{\text{crustal-Fe}}$, and ([d-Fe]/[d-Al])$_{\text{high-temp-FeOx-dFe}}$ are the molar [d-Fe]/[d-Al] ratios of aerosol particles,
crustal-Fe, and high-temp-FeOx, respectively. In addition, the $f_{\text{high-temp-FeOx-dFe}}$ of TSP (= coarse + fine aerosol particles) was
estimated by using the [d-Fe]/[d-Al] ratio of TSP. For the comparison of the $f_{\text{high-temp-FeOx-dFe}}$ of TSP, the relative abundance of

non-crustal d-Fe in fine aerosol particles to that of d-Fe in TSP samples ($f_{\text{high-temp-FeOx-dFe-fine/TSP}}$) was calculated by using the
following equation:

$$(f_{\text{high-temp-FeOx-dFe/TSP}}) = \Sigma(f_{\text{high-temp-FeOx-dFe}} \times [\text{d-Fe}])_{\text{fine}} / [\text{d-Fe}]_{\text{TSP}}, \text{ (Eq. 9)}$$

where $\Sigma(f_{\text{high-temp-FeOx-dFe}} \times [\text{d-Fe}])_{\text{fine}}$ is the summation of non-crustal d-Fe concentration in fine aerosol particles (<1.3 μm).
The $f_{\text{high-temp-FeOx-dFe/TSP}}$ calculated by using Equation 9 was almost identical to the $f_{\text{non-crustal-Fe}}$ of TSP calculated by using

Equations 7 and 8 (Fig. S7). This result provided evidence that the coarse aerosol particles contained low amounts of non-
crustal d-Fe that increased the [d-Fe]/[d-Al] ratio of TSP.

### 3.7.2.  The fraction of high-temp-FeOx to d-Fe

The calculation showed that the annual average of the fractions of non-crustal Fe in d-Fe in TSP was 19.9 % (Fig. 10a,

range: 1.48–80.7 %). The fraction of non-crustal Fe calculated with the [d-Fe]/[d-Al] ratio of 4.67 was lower than that
calculated with the [d-Fe]/[d-Al] ratio of 2.08 (Figs. S8a and S8b). High non-crustal Fe fractions were observed in summer
and fall (average: 29.4 ± 25.8 %, range: 9.41–80.7 %) when the air mass originated from the domestic region (Fig. S1). By
contrast, non-crustal Fe (average: 13.5 ± 10.6 %, range: 1.48–34.4 %) in winter and spring, when air masses originated from
East Asia, was lower than that in summer and fall. A satellite-based observation reported that approximately half of aerosol

transport events from East Asia to the Pacific Ocean between 2007 and 2016 occurred in spring and that dust-associated events
mainly occurred in spring (Zhu, Q., 2020). In addition, atmospheric Fe is supplied to the surface ocean by an episodic event
that accounts for approximately 30–90 % of annual Fe depositions within 5 % of the days of a year (Mahowald et al., 2009).
In consideration of these phenomena, mineral dust was the dominant source of d-Fe in aerosol particles deposited in the North





Pacific Ocean. Nevertheless, the contribution of high-temp-FeOx, as a source of d-Fe in surface seawater during dust events,

cannot be negligible because the average fraction of non-crustal Fe during spring and dust events was $10.2 \pm 7.58$ %.

On the basis of the fractions of crustal and non-crustal Fe, the $Fe_{sol}$% of crustal and non-crustal Fe in fine aerosol particles were estimated by using following equations:

$$\text{Crustal-Fe}_{sol}\% = [(\text{d-Fe} \times f_{\text{crustal-dFe}})/\text{crustal Fe}] \times 100, \text{ (Eq. 10)}$$

$$\text{High-temp-FeOx-Fe}_{sol}\% = [(\text{d-Fe} \times f_{\text{high-temp-FeOx-dFe}})/\text{non-crustal Fe}] \times 100. \text{ (Eq. 11)}$$

Crustal and non-crustal Fe concentrations were calculated by using Equations 3 and 4, respectively. The annual average crustal-$Fe_{sol}$% and non-crustal-$Fe_{sol}$% of fine aerosol particles were 14.1 % (0–43.7 %) and 9.35 % (0.501–46.2 %), respectively (Figs. 8c and 8d). The average crustal-$Fe_{sol}$% of TSP was $6.52 \pm 3.05$ % ([d-Fe]/[d-Al]: 2.08) because insoluble Fe in mineral dust was mainly distributed in coarse aerosol particles. When the [d-Fe]/[d-Al] ratio was set to 4.67, the crustal-$Fe_{sol}$% and non-crustal-$Fe_{sol}$% of fine aerosol particles were 20.2 % (0.0285–65.1 %) and 4.62 % (0.209–35.9 %), respectively (Figs. S8c and

S8d). The higher value of crustal-$Fe_{sol}$% when the [d-Fe]/[d-Al] ratio was set to 4.67 than that when the [d-Fe]/[d-Al] ratio was set to 2.08 was due to the large contribution of crustal Fe to d-Fe. Thus, when focusing on fine aerosol particles, mineral dust showed higher $Fe_{sol}$% than high-temp-FeOx. One reason is that high-temp-FeOx was derived from local anthropogenic emissions and may not have undergone considerable chemical aging. Another possible reason is that at pH 1.7 and 4.3, the $Fe_{sol}$% of submicron Fe oxides was lower than that of submicron illite and kaolinite (Marcotte et al. 2020). As previously

mentioned, fine aerosol particles with high $Fe_{sol}$% (>10 %) experienced highly acidic conditions (pH < 3.0). Therefore, the high $Fe_{sol}$% shown by mineral dust in non-crustal Fe in fine aerosol particles is consistent with the results of laboratory experiments (Marcotte et al. 2020). The $Fe_{sol}$% of fine aerosol particles in the 0.39–0.69 and 0.69–1.3 μm fractions were higher than that in the finest fraction throughout the year (Fig. 6c), indicating that mineral dust in the 0.39–0.69 and 0.69–1.3 μm fractions was more aged than that in the finest fraction. Single-particle analyses of aerosol particles in East Asia showed that

the abundance of the internally mixed particles of Fe-bearing particles with sulfate was the highest at approximately 0.7 μm of aerosol diameter (Sullivan et al., 2007a; Li et al., 2017; Zhu, Y. et al., 2020, 2022). In addition, $Fe_{sol}$% of mineral dust in the 0.39–0.69 and 0.69–1.3 μm fractions collected in the Pacific Ocean were higher than that in coarse aerosol particles due to chemical aging via condensation–evaporation cycle during transport (Sakata et al., 2022). Therefore, the chemical aging of mineral dust in fine aerosol particles plays an important role in the supply of Fe to the ocean surface.


## 4.    Implication for marine aerosol particles

The availability of the [d-Fe]/[d-Al] ratio for evaluating the emission sources of d-Fe in marine aerosol particles was investigated by using the observational results of previous studies (Buck et al., 2006, 2010b; Shelley et al., 2018; Baker et al., 2020; Sakata et al., 2022). In general, soluble metals in aerosol particles were extracted by using MQ water through

instantaneous or batch leaching, and only Baker et al. (2020) employed ammonium acetate solution with a pH of 4.7. The [d-Fe]/[d-Al] ratio of the marine aerosol samples collected in the Pacific and Atlantic Oceans were rarely higher than 1.0 (Fig.



11a and 11b). The fractions of non-crustal d-Fe in TSP collected from the Atlantic and Pacific Oceans were 9.58 % and 13.4 %, respectively (Fig. 11c). In the Pacific Ocean, the contribution of non-crustal d-Fe tends to be higher in the region east of 170 °E because anthropogenic Fe in fine aerosol particles is transported farther than that in coarse mineral dust particles (Fig.

12a, Mahowald et al., 2018). By contrast, the large fraction of anthropogenic Fe in the Atlantic was found around the coastal area in North America and Europe (Fig. 12b). This result was consistent with the estimation of the non-crustal Fe fraction based on $\delta^{56}$Fe, which indicated that more than half of the d-Fe in Fe around the coastal regions was derived from anthropogenic Fe with a negative $\delta^{56}$Fe (Conway et al., 2019). However, the fraction of non-crustal Fe in TSP with negative $\delta^{56}$Fe values estimated on the basis of the [d-Fe]/[d-Al] ratio was only 12.4 ± 2.92 % (Fig. 11c). Thus, a discrepancy existed between the

results of d-Fe emission source estimates based on Fe isotopic ratios and those based on [d-Fe]/[d-Al]. One of the reasons is for this discrepancy is the limited data on the [d-Fe]/[d-Al] and $\delta^{56}$Fe ratios of emission source samples, in particular anthropogenic emissions. The [d-Fe]/[d-Al] ratio and $\delta^{56}$Fe of high-temp-FeOx were estimated on the basis of the measurement values of aerosol particles. The [d-Fe]/[d-Al] ratios of mineral dust and non-magnetic aluminosilicates of coal combustion and MSWI had a small effect on the uncertainty of the non-crustal Fe fraction estimates because measured values were available.

Therefore, further studies involving the accumulated data on the [d-Fe]/[d-Al] ratio and $\delta^{56}$Fe of high-temp-FeOx collected from various industrial sites are required. Currently, the accurate $\delta^{56}$Fe and [d-Fe]/[d-Al] ratio of high-temp-FeOx may be obtained through the separation of magnetic particles as performed by Zuo et al. (2022). The accumulated data on the previously reported values of [d-Fe]/[d-Al] can be used and are expected to provide insight into the time-series variation in the contribution of anthropogenic Fe to aerosols in marine aerosols.

560         Figure 11b shows a scatter plot of the [d-Fe]/[d-Al] ratios and Fe$_{sol}$% of size-fractionated aerosol particles collected from the Atlantic and Pacific Oceans (Baker et al., 2020; Sakata et al., 2022). The average [d-Fe]/[d-Al] ratios of coarse and fine aerosol particles collected offshore of the Sahara Desert were 0.216 ± 0.163 and 0.155 ± 0.0549, respectively. These values were similar to the [d-Fe]/[d-Al] ratio of Saharan dust (0.108 ± 0.0609), indicating that d-Fe in these size-fractionated aerosol samples originated from Saharan dust. Iron in the size-fractionated aerosol particles collected in the western Pacific Ocean

was also derived from mineral dust regardless of aerosol diameter because the EF of Fe in all size fractions was almost 1 (Sakata et al., 2022). The average [d-Fe]/[d-Al] ratio of coarse aerosol particles collected above the Pacific Ocean (0.378 ± 0.104) was slightly higher than the average [d-Fe]/[d-Al] ratio of Asian dust (0.238 ± 0.201) but was similar to the ratio of coarse aerosol particles collected in this study (0.408 ± 0.168). Therefore, d-Fe in these samples was derived from the hydrolysis or proton-promoted dissolution of mineral dust. Although Fe in coarse aerosol particles collected in the Atlantic

and Pacific Oceans was derived from mineral dust, the [d-Fe]/[d-Al] ratio of the coarse and fine aerosol particles above the Atlantic Ocean was lower than that of coarse aerosol particles collected in the Pacific Ocean. The differences in the mineralogical compositions of mineral dust in the hinterlands account for the differences in the ratios of the aerosol particles collected in the Atlantic and Pacific Oceans. Therefore, measuring the [d-Fe]/[d-Al] ratio of soil samples around the sampling site is important to determine the representative [d-Fe]/[d-Al] ratio of mineral particles in aerosol particles.



The high Fe$_{sol}$% (>10%) of fine aerosol particles in the Pacific was attributed to ferric organic complexes of humic-like substances (Fe(III)-HULIS, Sakata et al. 2022). The [d-Fe]/[d-Al] ratio of fine aerosol particles containing Fe(III)-HULIS is expected to be between 1.0 and 1.5 due to ligand-promoted dissolution. However, the [d-Fe]/[d-Al] ratio of fine aerosol particles containing Fe(III)-HULIS was $0.440 \pm 0.117$ (range: 0.255–0.567), which was consistent with the ratio of Asian dust that underwent proton-promoted dissolution. A previous study reported that the Fe$_{sol}$% of these samples was enhanced by the

aerosol acidification of aluminosilicates; Fe(III)-HULIS then formed via complexation reactions in cloud water (Sakata et al., 2022). Thus, the [d-Fe]/[d-Al] ratio may not change even if the Fe species was altered by atmospheric processes after Fe solubilization. However, this point needs further investigation. The determination of the [d-Fe]/[d-Al] ratio via MQ extraction without organic ligands (e.g., oxalate and acetate) is preferred for investigating the contribution of ligand-promoted dissolution to Fe$_{sol}$% enhancement because the ratio is altered if the extracted solution contains organic ligands (e.g., coarse aerosol [P]

vs. coarse aerosol [L], Fig. 9a). However, data on the [d-Fe]/[d-Al] ratio of size-fractionated marine aerosol particles under the effect of high-temp-FeOx have not yet been published at this time. Therefore, further studies on the [d-Fe]/[d-Al] ratio of marine aerosol particles influenced by high-temp-FeOx are required. If future studies detect high [d-Fe]/[d-Al] ratios in samples that are strongly affected by high-temp-FeOx, the [d-Fe]/[d-Al] ratio will be a powerful tool for estimating the fraction of crustal and non-crustal Fe in d-Fe in marine aerosol particles.


## 5.   Conclusion

Annual observations were conducted on total and dissolved Fe and Al concentrations in size-fractionated aerosol particles. Total Fe and Al concentrations were mainly distributed in coarse aerosol particles, whereas d-Fe and d-Al concentrations dominated in fine aerosol particles. Considering the higher d-Fe concentration and longer residence time of fine aerosol

particles than those of coarse aerosol particles, the atmospheric deposition of fine aerosol particles was the dominant source of d-Fe in the surface ocean. Since the [d-Fe]/[d-Al] ratio of coarse aerosol particles ($0.408 \pm 0.168$) was lower than that of fine aerosol particles ($1.15 \pm 0.803$), the sources of d-Fe differed between coarse and fine aerosol particles. The [d-Fe]/[d-Al] ratio of coarse aerosol particles was similar to that observed of Fe dissolved via proton-promoted dissolution. Indeed, Fe$_{sol}$% increased as the surfaces of coarse particles became hygroscopic through reaction with sulfuric acid. This effect intensified

with the reduction in particle size. The fine aerosol particles presented a wide range of [d-Fe]/[d-Al] ratios (0.386–4.67). The correlation of the [d-Fe]/[d-Al] ratio of the fine aerosol particles with the EF of Fe indicated that high [d-Fe]/[d-Al] ratios (>1.5) could be attributed to non-crustal Fe. High-temp FeOx in magnetic particles was likely the dominant species in non-crustal Fe with high [d-Fe]/[d-Al] ratios. The fractions of mineral dust and high-temp-FeOx were evaluated on the basis of the [d-Fe]/[d-Al] ratios and emission sources of aerosol particles. Approximately 80 % of d-Fe in TSP collected in spring, when

numerous events of aerosol transport from East Asia to the North Pacific occurred, originated from mineral particles. The high fraction of mineral dust was attributed to the higher Fe$_{sol}$% of crustal Fe than that of high-temp-FeOx. The Fe$_{sol}$% of submicron mineral dust (e.g., illite) was known to be higher than that of hematite and magnetite of the same size under acidic conditions. Furthermore, the Fe$_{sol}$% of fine particles increased with the decrease in aerosol pH, indicating that the acidification of mineral





particles played an important role in the supply of d-Fe to surface seawater in the North Pacific via atmospheric deposition.

Thus, the source estimation of d-Fe by using the [d-Fe]/[d-Al] ratio is a powerful tool for estimating not only the source of d-Fe in marine aerosols but also the dissolution processes of Fe in aerosol particles. The source estimation of d-Fe by using the [d-Fe]/[d-Al] ratio also has the advantage of being performed through a simple analytical method in which Fe and Al are extracted by using MQ and their concentrations are measured through ICP–MS. Therefore, this method can be easily conducted not only in future studies but also on aerosol samples collected in previous studies. Currently, data on the [d-Fe]/[d-Al] ratios

of aerosols that are strongly affected by anthropogenic Fe and emission sources associated with high-temperature combustion processes are limited. Therefore, collecting data on the [d-Fe]/[d-Al] ratios of these samples will enable us to identify the emission sources of d-Fe in marine aerosol particles with increased robustness.

***Data Availability*** All quantitative data will be available at the ERAN database (https://www.ied.tsukuba.ac.jp/database/00156.html) with a doi: 10.34355/CRiED.U.Tsukuba.00156.

***Author contributions***. The study was designed by K.S., A.S., and Y.T. Aerosol sampling was performed by K.S. and Y.Y. Major ion concentrations were determined by K.S., Y.Y., and C.M. Trace metal concentrations were measured by K.S., C.M., and M.K. The paper was written by K.S., and Y.T., and all authors were reviewed the manuscript before submission.

***Competing interests***. The authors declare that they have no conflict of interest.

***Financial support.*** The studies were supported by Grant-in-Aid for JSPS Research Fellow (14J06437).





**Figure captions**

Figure 1: Sites (Higashi-Hiroshima) of size-fractionated aerosol sampling. Maps were visualized by Ocean Data View (Schlitzer, 2021).

Figure 2: Monthly variations and size distributions of (a) $Na^+$, (b) $Ca^{2+}$, (c) $Mg^{2+}$, (d) $K^+$, (e) $NH_4^+$, (f) $Cl^-$, (g) $NO_3^-$, (h) $SO_4^{2-}$, and (i) nss-$SO_4^{2-}$. The summation of all fractions corresponds to TSP concentration.

Figure 3: Monthly variations and size distributions of (a) total Al and (b) total Fe concentrations. (c) Box plot of $Fe_{sol}$% in each size fraction.

Figure 4: Scatter plots of non-crustal Fe concentration with (a) Cu, (b) Zn, (c) Sb, and (d) Pb in coarse aerosol particles.

Figure 5: Scatter plots of non-crustal Fe concentration with (a) Zn, (b) Sb, (c) Pb, and (d) V in fine aerosol particles.

Figure 6: Monthly variation and size distributions of (a) d-Fe and (b) $Fe_{sol}$%. (c) Box plot of $Fe_{sol}$% in each size fraction. Monthly variation and size distributions of (d) d-Al and (e) $Al_{sol}$%. (f) Box plot of $Al_{sol}$% in each size fraction.

Figure 7: Scatter plots of $Fe_{sol}$% and $Al_{sol}$% in coarse aerosol particles with (a) nss-$SO_4^{2-}$/Fe or Al and (b) $NO_3^-$/Fe or Al.

Scatter plots of $Fe_{sol}$% and $Al_{sol}$% in fine aerosol particles with (c) nss-$SO_4^{2-}$/Fe or Al, (d) $NO_3^-$/Fe or Al, and (e) aerosol pH.

Figure 8: Scatter plots of d-Fe with (a) non-crustal Fe, (b) total V, (c) total Cu, (d) total Sb, and (e) total Pb concentrations in fine aerosol particles.

Figure 9: (a) Box plot of [d-Fe]/[d-Al] in emission source samples [P] and [L] indicating proton-promoted and ligand-promoted

dissolutions. Scatter plots of [d-Fe]/[d-Al] ratio with (b) the $Fe_{sol}$% and (c) EF of Fe. Light blue circles and black squares in panels (b) and (c) represent the data on coarse and fine aerosol particles collected in Higashi-Hiroshima, respectively. Pink, yellow, light green, and gray regions show the typical ranges of the [d-Fe]/[d-Al] ratios of aluminosilicate glasses of coal combustion and MSWI, the proton- and ligand-promoted dissolution of mineral dust, and high-temp-FeOx, respectively. The regions were decided on the basis of the box plots of panels (a).

Figure 10: Monthly variations of (a) non-crustal Fe fraction in TSP and (b) non-crustal Fe fraction in fine aerosol particles, (c) crustal-$Fe_{sol}$%, and (d) non-crustal $Fe_{sol}$% when the [d-Fe]/[d-Al] ratio of non-crustal Fe is 2.08.

Figure 11: Scatter plots of the [d-Fe]/[d-Al] ratio with $Fe_{sol}$% in (a) TSP samples and (b) size-fractionated aerosol particles collected in the marine atmosphere. (c) The fraction of non-crustal d-Fe in these samples was calculated by using Equations 6 and 7. The [d-Fe]/[d-Al] ratios of mineral dust in the Pacific and Atlantic Oceans were the average

ratios of Asian dust (0.238) and Saharan dust (0.163), respectively. The [d-Fe]/[d-Al] ratio of non-crustal Fe was fixed at 2.08. The [d-Fe]/[d-Al] ratios of TSP and size-fractionated aerosol samples were adapted from Buck et al. (2006, 2010b), Shelley et al. (2018), Baker et al. (2020), and Sakata et al. (2022). The Fe isotopic ratio of TSP in the Atlantic Ocean was reported by Conway et al. (2019). Pink, yellow, light green, and gray regions in panels (a) and (b) show the typical [d-Fe]/[d-Al] ratios of coal/MSWI fly ash, proton-promoted and ligand-promoted mineral

dust dissolution, and pyrogenic Fe oxides, respectively.





Figure 12: Non-crustal Fe fractions in TSP samples collected from (a) the Pacific Ocean and (b) Atlantic Ocean. Non-crustal Fe fractions were calculated by using reported [d-Fe]/[d-Al] ratios.



**Figures**



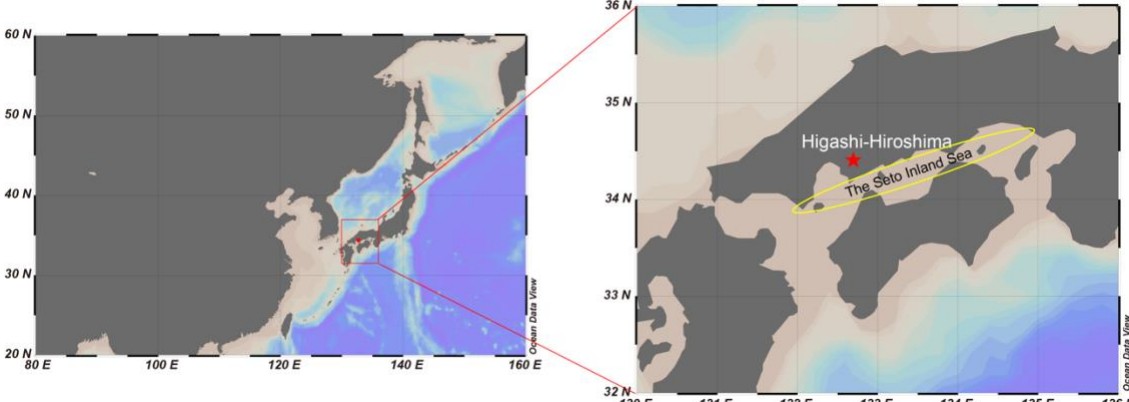

Figure 1: Sites (Higashi-Hiroshima) of size-fractionated aerosol sampling. Maps were visualized by Ocean Data View (Schlitzer, 2021).







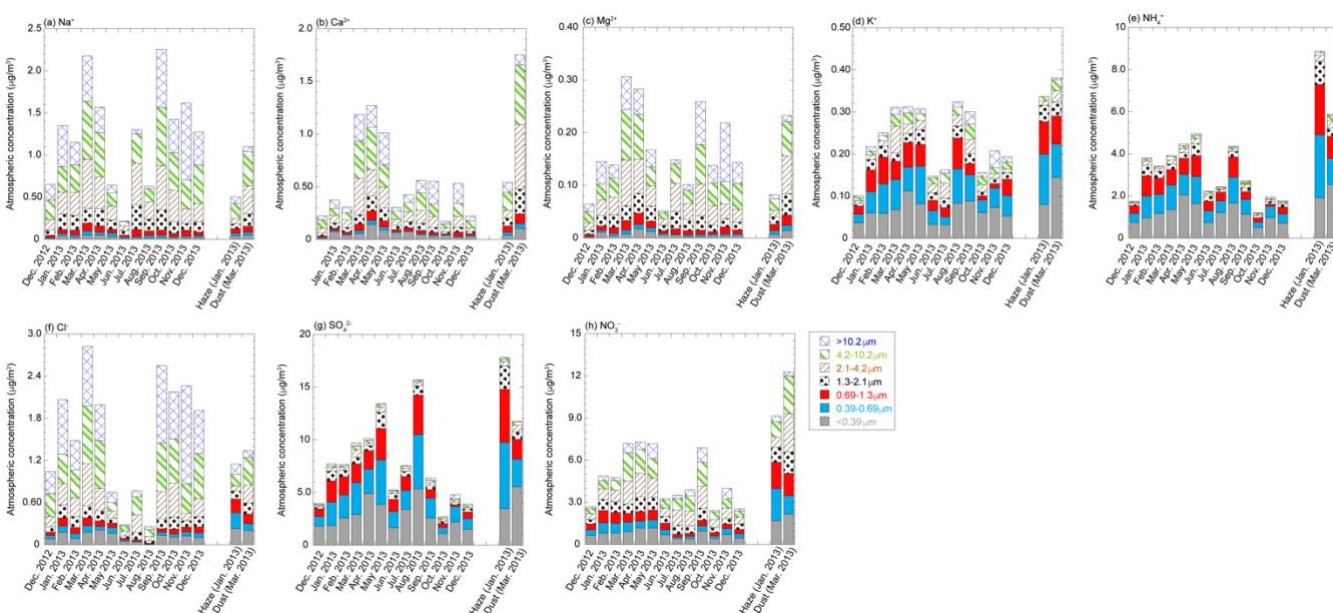


Figure 2: Monthly variations and size distributions of (a) Na$^+$, (b) Ca$^{2+}$, (c) Mg$^{2+}$, (d) K$^+$, (e) NH$_4^+$, (f) Cl$^-$, (g) NO$_3^-$, (h) SO$_4^{2-}$, and (i) nss-SO$_4^{2-}$. The summation of all fractions corresponds to TSP concentration.







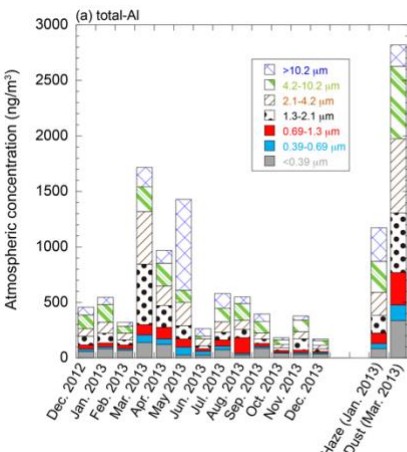 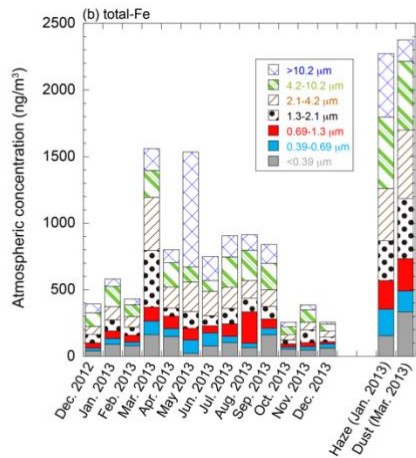 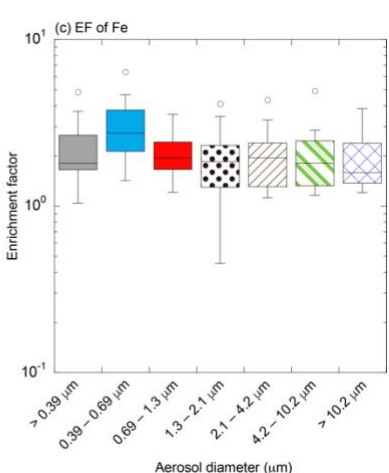

Figure 3: Monthly variations and size distributions of (a) total Al and (b) total Fe concentrations. (c) Box plot of Fe$_{sol}$% in each
size fraction.






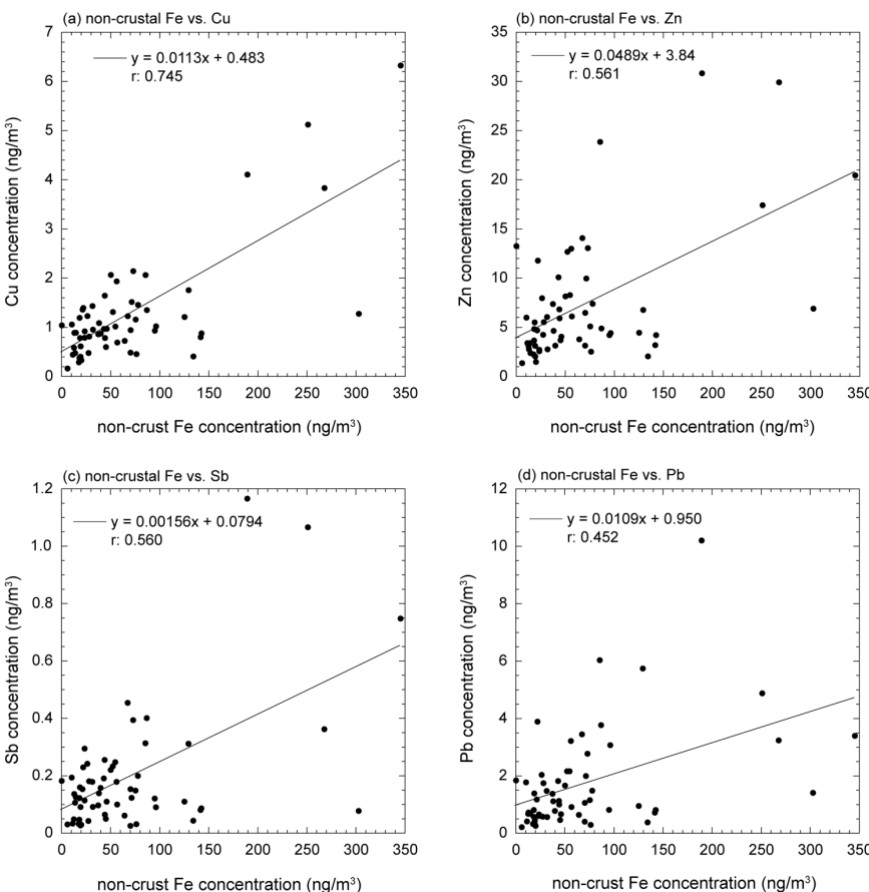


Figure 4: Scatter plots of non-crustal Fe concentration with (a) Cu, (b) Zn, (c) Sb, and (d) Pb in coarse aerosol particles.




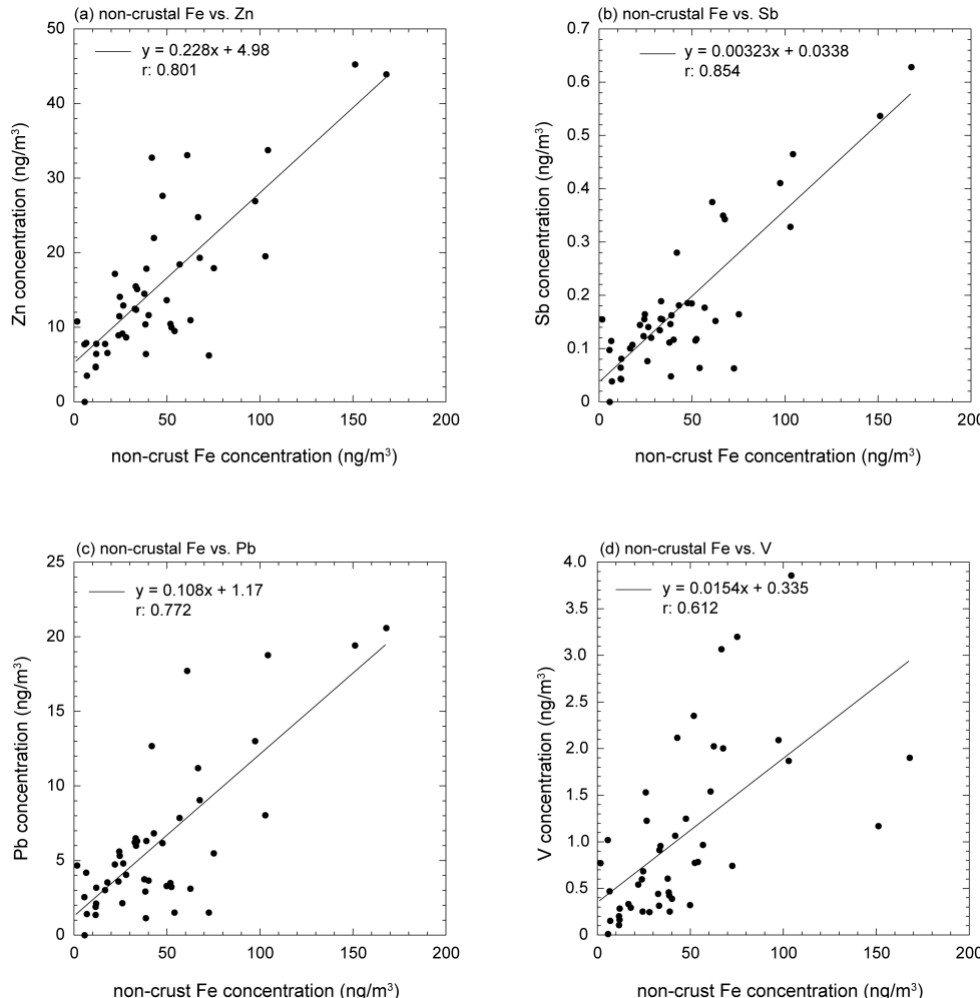

Figure 5: Scatter plots of non-crustal Fe concentration with (a) Zn, (b) Sb, (c) Pb, and (d) V in fine aerosol particles.








Figure 6: Monthly variation and size distributions of (a) d-Fe and (b) $Fe_{sol}$%. (c) Box plot of $Fe_{sol}$% in each size fraction. Monthly variation and size distributions of (d) d-Al and (e) $Al_{sol}$%. (f) Box plot of $Al_{sol}$% in each size fraction.









Figure 7: Scatter plots of $Fe_{sol}$% and $Al_{sol}$% in coarse aerosol particles with (a) nss-$SO_4^{2-}$/Fe or Al and (b) $NO_3^-$/Fe or Al.

750       Scatter plots of $Fe_{sol}$% and $Al_{sol}$% in fine aerosol particles with (c) nss-$SO_4^{2-}$/Fe or Al, (d) $NO_3^-$/Fe or Al, and (e) aerosol pH.








Figure 8: Scatter plots of d-Fe with (a) non-crustal Fe, (b) total V, (c) total Cu, (d) total Sb, and (e) total Pb concentrations in

760          fine aerosol particles.

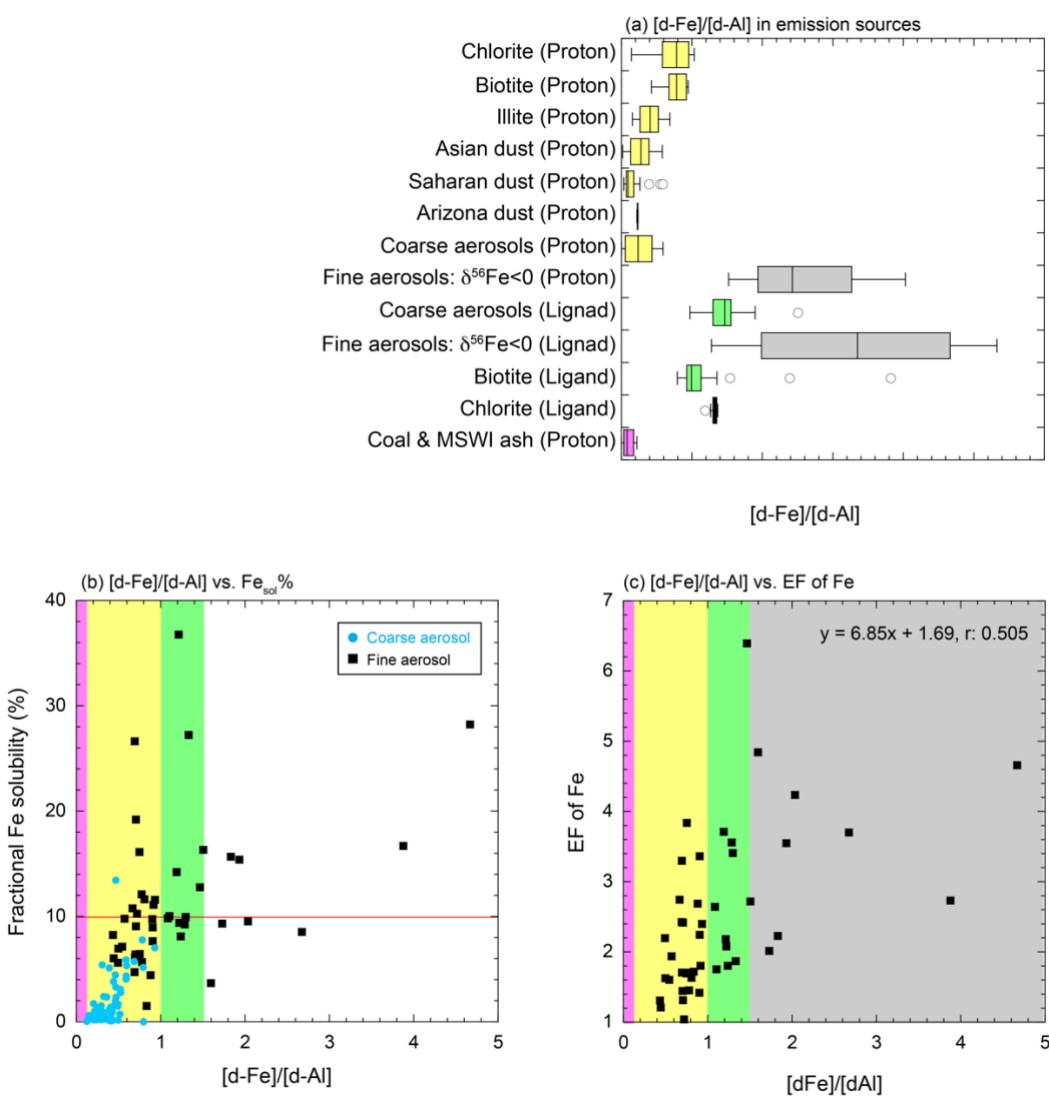


Figure 9: (a) Box plot of [d-Fe]/[d-Al] in emission source samples [P] and [L] indicating proton-promoted and ligand-promoted dissolutions. Scatter plots of [d-Fe]/[d-Al] ratio with (b) the $Fe_{sol}$% and (c) EF of Fe. Light blue circles and black squares in panels (b) and (c) represent the data on coarse and fine aerosol particles collected in Higashi-Hiroshima, respectively. Pink, yellow, light green, and gray regions show the typical ranges of the [d-Fe]/[d-Al] ratios of aluminosilicate glasses of coal combustion and MSWI, the proton- and ligand-promoted dissolution of mineral dust, and high-temp-FeOx, respectively. The regions were decided on the basis of the box plots of panels (a).






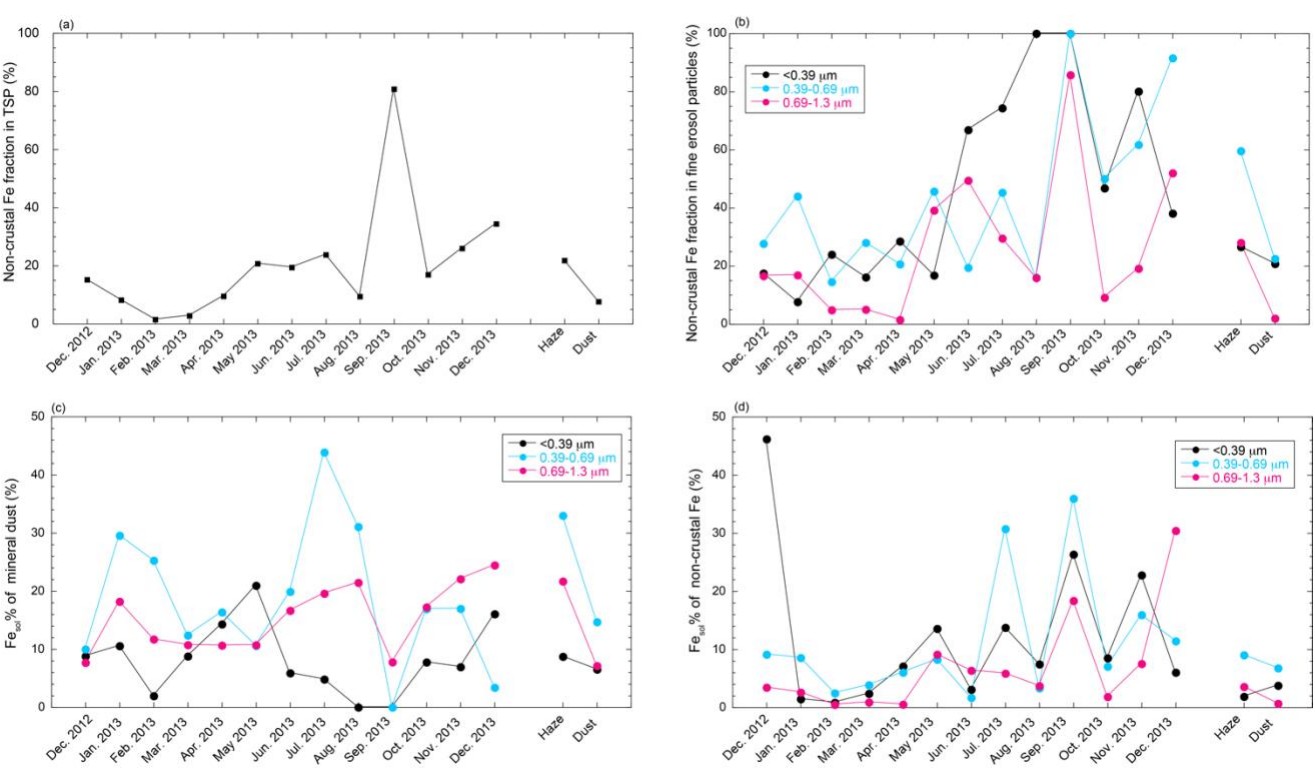

Figure 10: Monthly variations of (a) non-crustal Fe fraction in TSP and (b) non-crustal Fe fraction in fine aerosol particles, (c) crustal-Fe$_{sol}$%, and (d) non-crustal Fe$_{sol}$% when the [d-Fe]/[d-Al] ratio of non-crustal Fe is 2.08.






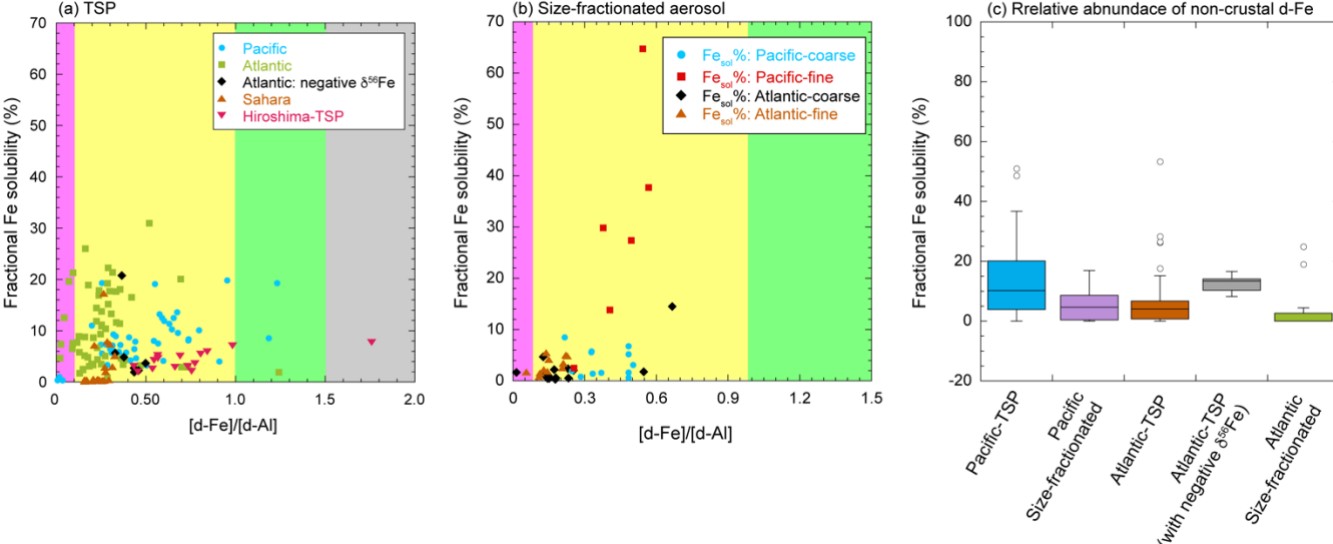

Figure 11: Scatter plots of the [d-Fe]/[d-Al] ratio with Fe$_{sol}$% in (a) TSP samples and (b) size-fractionated aerosol particles collected in the marine atmosphere. (c) The fraction of non-crustal d-Fe in these samples was calculated by using Equations 6 and 7. The [d-Fe]/[d-Al] ratios of mineral dust in the Pacific and Atlantic Oceans were the average ratios of Asian dust (0.238) and Saharan dust (0.163), respectively. The [d-Fe]/[d-Al] ratio of non-crustal Fe was fixed at 2.08. The [d-Fe]/[d-Al] ratios of TSP and size-fractionated aerosol samples were adapted from Buck et al. (2006, 2010b), Shelley et al. (2018), Baker et al. (2020), and Sakata et al. (2022). The Fe isotopic ratio of TSP in the Atlantic Ocean was reported by Conway et al. (2019). Pink, yellow, light green, and gray regions in panels (a) and (b) show the typical [d-Fe]/[d-Al] ratios of coal/MSWI fly ash, proton-promoted and ligand-promoted mineral dust dissolution, and pyrogenic Fe oxides, respectively.



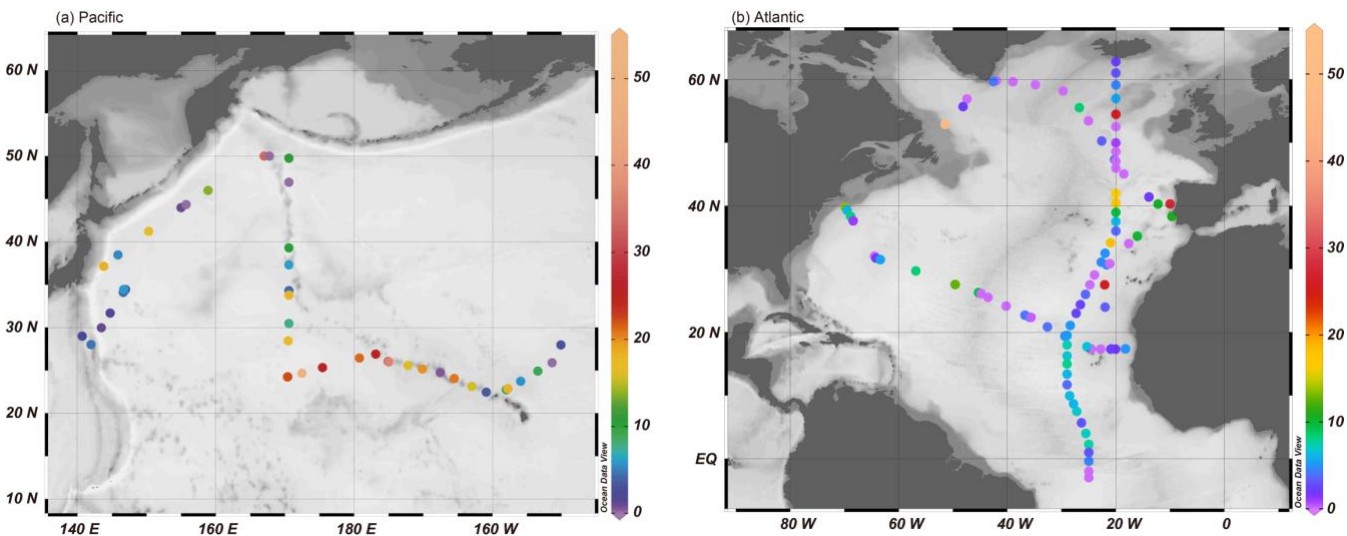

Figure 12: Non-crustal Fe fractions in TSP samples collected from (a) the Pacific Ocean and (b) Atlantic Ocean. Non-crustal
        Fe fractions were calculated by using reported [d-Fe]/[d-Al] ratios reported by Buck et al. (2006, 2010b) and
        Shelley et al. (2018). The figures were described by Ocean Data View (Schlitzer, 2021).





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
