# Peer review of "Measurement report: Stoichiometry of dissolved iron and aluminum as an indicator of the factors controlling the fractional solubility of aerosol iron: Results of the annual observations of size-fractionated aerosol particles in Japan"

_Atmospheric Chemistry and Physics, 2023_

## Author Comment (AC2)

Reviewer 1

| | Comments | Reply |
|---|---|---|
| 1.1. | Aerosol Fe solubility is a key parameter for impacts of aerosol Fe on marine biogeochemistry, is still not well constrained. Several sources and processes may contribute to dissolved aerosol Fe, but quantitative explanations are still difficult. Sakata et al. explored the possibility to use the ratio of dissolved Al to dissolved Fe to understand factors which control aerosol Fe solubility in fine and coarse particles, and then discussed sources of dissolved aerosol Fe. This idea is novel, and the results are very interesting. This manuscript can still be substantially improved to increase its readability, clarity and impacts. I also posted a community comment 28 March, and the authors may also need to take it into account.

Overall, I urge the authors to check this manuscript carefully and thoroughly, as there are a lot of language issues. Professional editing is also recommended. In the community comment I posted on 28 March, I provided a few examples. | We sincerely thank the reviewer for the time and effort put into this review. We have carefully revised the manuscript with full consideration of the comments and suggestions provided. We also reply to your comments posted as community comments in this sheet. Please find the point-to-point replies listed below. We apologize for the inconvenience for your peer-preview processes due to our poor English. English editing was conducted by a native English speaker prior to submission.
"Revised text as it appears in the text (in quotes, blue font)" |
| 1.2. | Line 15-17: The second sentence in the abstract largely repeats the first sentence. In addition, there are some typos elsewhere in this manuscript (for example, line 305-306; line 29: should there be "for fine particles" after "from 0.386 to 4.67"?). The authors may want to check the language thoroughly. | Thank you for pointing out. We have removed the first sentence in abstract. |
| 1.3. | Line 155-159: It would be nice to provide values used for [X]/[Na+] for seawater and references. | The molar ratios of target ions relative to Na+ have been added in the manuscript.
The molar ratios of $[K^+]/[Na^+]$, $[Mg^{2+}]/[Na^+]$, $[Ca^{2+}]/[Na^+]$, and $[SO_4^{2-}]/[Na^+]$ were 0.0213, 0.113, 0.0213, and 0.0596, respectively (Nozaki, 2001). |
| 1.4. | Line 267-268: Although these numbers can be found in the SI, could the author provide average EF values of Fe in TSP for different seasons? | Thank you for the suggestions. We provided EF values in TSP derived from the Japanese atmosphere (in summer and fall) and East Asia.
The average EF of Fe in the TSP samples collected from summer to fall (June to October) was $2.84 \pm 0.83$ when air masses mainly originated from Japan (Fig. S6a). By contrast, the average EF of Fe in the TSP derived from East Asia in winter and spring was $1.57 \pm 0.35$ (Fig. S6a). Thus, the influences of |

| | | |
|---|---|---|
| | | anthropogenic emission on the total Fe in TSP are more significant in the air mass derived from Japan than that from East Asia (Figs. S1, S2, and S6a). |
| 1.5. | Line 305-306: This sentence "The d-Fe and d-Al of fine aerosol particles were 72.0 ± 8.87 % and 53.1 ± 9.90 %, respectively" may need revision. | Thank you for pointing out. The sentence has been improved as below. The average d-Fe and d-Al concentrations of fine aerosol particles were 27.3 ± 17.4 and 14.3 ± 10.9 ng m-3, which accounted for 72.0 ± 8.9 % and 53.1 ± 9.9 % of d-Fe and d-Al in the TSP, respectively. |
| 1.6. | Line 311-318: Our previous work (Zhang et al., 2022; Zhang et al., 2023) also found and try to explain enrichment of dissolved Fe in fine particles, when compared to total Fe. Recently we published two papers (Zhang et al., 2022; Zhang et al., 2023), which discussed sources of dissolved Fe and Fe solubility for fine and coarse particles and are highly relevant to the work presented by Sakata et al. Our two papers reported many similar results as Sakata et al., including what Sakata et al. presented in Sections 3.3-3.4. There are also some differences between our work and the work by Sakata et al. (2023). Therefore, I would like to bring the authors' attention to our recent work (Zhang et al., 2022; Zhang et al., 2023). Section 3.4.2: Our previous studies (Zhang et al., 2022; Zhang et al., 2023) also suggested that chemical aging played a more important role in regulating Fe solubility in coarse particles than fine particles. | These three comments are similar and have been combined into one. Thank you for bringing our attention to the two papers. We briefly compared the result between our study and your previous studies, and the following sentence was added to the manuscript. These results are reasonable and supported by previous studies reporting that mineral dust and anthropogenic Fe in fine aerosol particles were covered by sulfate, and internal mixing of these particles with sulfate promoted the dissolution of Fe (Sullivan et al., 2007; Fitzgerald et al., 2015; Li et al., 2017, Zhu, Y. et al., 2020, 2022). However, compared with the correlation factor of Fesol% with [nss-SO4$^{2-}$]/[total Fe], coarse aerosol particles yielded a higher correlation factor than fine aerosol particles (Figs. 4a and 4c). Similar results have been reported by previous studies on Fesol% in China, which indicated the presence of anthropogenic Fe with high Fesol% as one of the reasons for the low value of the correlation factor (Zhang et al., 2022, 2023). Therefore, the low correlation between Fesol% and [nss-SO4$^{2-}$]/[total Fe] of fine aerosol particles implied that acidification of mineral dust was not the sole factor controlling Fesol%. There are several different points in the results between this and your studies, but we did not discuss the reasons. This is because (i) it is not easy to compare results between this and your studies due to many differences in the sampling methods (e.g., stage numbers of impactors, sampling periods, and date), and (ii) detailed comparisons between these studies would make this paper longer and consequently conflict with your other comment (No. 1.7). |

| | | |
|---|---|---|
| | | The difference between our and your studies would be whether $Fe_{sol}$% is correlated with $[NO_3^-]/[total\text{-}Fe]$ or not. In the case of coarse aerosol particles, we believed that $HNO_3$ mainly reacted with $CaCO_3$ in mineral dust as described in Supplemental Note. In the case of fine aerosol particles, no relationship between the $Fe_{sol}$% and $[NO_3^-]/[total\text{-}Fe]$ seems to be found because $NO_3^-$ in aerosols prefers the vapor phase due to its low solubility in water under high temperature and acidic conditions. Indeed, Zhang et al. (2023) have reported that the correlation between $Fe_{sol}$% and $[NO_3^-]/[total\text{-}Fe]$ in summer was worse. Therefore, the nitrate evaporation associated with pH cycling during aerosol transport is likely the reason for no correlation between $[NO_3^-]/[total\text{-}Fe]$, but further research is needed. |
| 1.7. | It may increase the readability to group Sections 3.5-3.7 into a new section. The current Section 3 is very long, and contents in Sections 3.5-3.7 are different from Section 3.1-3.4? In addition, I feel the current manuscript is very long, and the quality of some figures is not good (they contain many panels and are very busy). Sections 3.5-3.7 contain key information in this manuscript, but are not easy to follow. Perhaps the authors could improve them during the revision, but I cannot provide specific suggestions. | Sections 3.5-3.7 of the first version were rearranged into a new section, Section 4 (title: Establishing the source estimation method for d-Fe using [d-Fe]/[d-Al]) in the revised version. This section has been improved to increase the readability. Regarding the paper length, the section on the ion concentration and several figures were moved to Supplemental Information. We hope you could understand our decision that we did not reduce the text length for the other sections to avoid ambiguity in the significance of the newly established indicators for source apportionment of d-Fe. |

---

## Author Comment (AC3)

Reviewer 2

| | Comments | Reply |
|---|---|---|
| 2.1. | This paper by Sakata et al. presented a detailed analysis of annual observations of Fesol% in size-fractionated (seven fractions) aerosol particles at Higashi-Hiroshima, Japan. The [d-Fe]/[d-Al] ratios were proposed to identify the source of d-Fe in aerosol particles and to understand the seasonal variability of the fraction of mineral dust and anthropogenic Fe in d-Fe in aerosols. They compared the differences of total Fe and dissolved Fe and [d-Fe]/[d-Al] between in coarse aerosol particles and fine aerosol particles. What's the most important, they provided a simple but useful marker ([d-Fe]/[d-Al] ratio) to estimate the emission sources of d-Fe in marine aerosol particles. Overall, this paper is well written and logical, and the scientific questions discussed clearly, which meets the scope of ACP. I recommend this manuscript to be published after the following comments are addressed. | We sincerely thank the reviewer for their time and effort on this review. We have carefully revised the manuscript with full consideration of the comments and suggestions provided. Please find the point-to-point replies listed below. "Revised text is shown in blue (in quotes, blue font) in this replies." |
| 2.2. | Line 15-17: The first two sentences in the Abstract seem to be repetitive and appear to be a mess. Please simplify the expression to make it clearer. | Thank you for pointing this out. We have removed the first sentence in the previous version. |
| 2.3. | Line 103-106: "Our previous study identified that fine aerosol particles collected at the sampling site contained anthropogenic Fe with a negative $\delta 56Fe$. Therefore, the sampling site is useful for evaluating the availability of the [d-Fe]/[d-Al] ratio as an indicator of the fractions of mineral dust and anthropogenic Fe in d-Fe in aerosol particles." Here the author stated that sampling location is of significance, however, in my opinion, there is no causal relationship between the preceding and following of this sentence. Why are sampling locations useful? Is it the unique research data provided? | We are sorry for the confusing descriptions. We intended to mention that Fe in size-fractionated aerosol particles at the sampling site is likely influenced by various emission sources, including mineral dust and anthropogenic emissions. We have improved the sentence as described below. Air masses at the sampling site in summer were mainly derived from the domestic region of Japan, whereas air masses passing over East Asia arrived at the site in winter and spring (Fig. S1). Our sample set included samples affected by Asian dust and serious haze events associated with anthropogenic emissions, which allowed us to obtain aerosol samples with considerable differences depending on the size dependence and seasonal variation expected for Fe chemistry and d-Fe sources. |
| 2.4. | Line 160: In the section of "Estimation of aerosol pH", as far as I know, the input data of E-AIM model requires high relative humidity, generally over 60%, so the authors | Since the observation site is under a humid environment throughout the year, the average RH for each sampling period exceeded 60% for most of the samples. The |

| | | | |
|---|---|---|---|
| | need to specify whether the data used meet the input requirements of the model. | following texts about the average RH during the sampling have been added to the manuscript.

In addition, E-AIM model IV cannot calculate aerosol pH when the RH is below 60 %. The average RH during each sampling period was higher than 60 %, except for those of aerosol samples collected in April and May 2013. The aerosol pH collected in April and May was calculated under the assumption of 60 % RH because the average RHs of the samples for these months were 59.4 % and 59.5 %, respectively. | |
| 2.5. | Normally, an enrichment factor greater than 10.0 for an element is generally considered to be an enrichment, possibly from anthropogenic source. However, it can be seen from Figure 3c that the EFs of Fe is less than 10.0 for all particle size, making it difficult to say that Fe was enriched. How do the authors interpret this? | As you pointed out, significant enrichment of target elements from anthropogenic emissions is usually recognized when EF is greater than 10. However, in this criterion, the contribution of anthropogenic Fe may be underestimated due to the smaller emission amount of anthropogenic Fe than mineral dust. Here, we evaluated the variability of the Fe/Al ratio in Asian dust based on previous studies. As a result, we found a small variation in the Fe/Al ratio in Asian dust (Fe/Al: $0.570\pm0.163$ (= $1\sigma$), range: 0.294–1.05). Therefore, we decided that enrichment of anthropogenic Fe is recognized when the EF of Fe is higher than 2 with consideration of small variability of the Fe/Al ratio in crustal materials. We have added the following sentences to the manuscript.

Iron and Al concentrations in the average continental crust (Fe/Al: 0.684) were acquired by referring to Taylor (1964). Given the variability of the Fe/Al ratio in crustal materials, significant enrichment of the Fe derived from anthropogenic emissions is usually recognized at EF values higher than 10.0. The EF equation suggests that about 90 % of Fe is derived from anthropogenic sources when the EF is 10.0. Given that the emission amount of crustal Fe is an order of magnitude higher than that of anthropogenic Fe, the EF for Fe in aerosol particles is usually below 10.0, except for aerosol samples collected near steel plants and in urban areas. Therefore, classification of Fe as anthropogenic Fe by the criterion EF > 10.0 substantially simplifies the origin of Fe in aerosol particles. If the variation of Fe/Al ratio in natural-source aerosol is limited in a | |

| | | |
|---|---|---|
| | | narrow range, aerosols with EF > 2.00 can still be evaluated as aerosol samples containing anthropogenic Fe component to a certain degree. The small variability of the Fe/Al ratio in desert soil in East Asia was confirmed (average ± 1σ standard deviation (ave.±1σ): 0.555 ± 0.170, range: 0.294–1.05, Nishikawa et al., 2013; Ding et al., 2001; Cao et al., 2008; Liu, X. et al., 2022 and references therein). The Fe/Al ratio in mineral dust exhibits a small variability, and thus, enrichment of anthropogenic Fe is recognized when the EF of Fe is higher than 2.00 (Fe/Al > 1.37). |
| 2.6. | Line 387: The statement of "The [d-Fe]/[d-Al] ratio is also decreased with increasing pH" is repeated. | Thank you for pointing out. We have revised it. |
| 2.7. | Too many figures, the authors can use correlation matrix to illustrate the relationship between Fe and other elements by merging Figure 4 and Figure 5. | We have rearranged the figures in the text according to your suggestion. Correlation matrixes associated with Figs. 4 and 5 are shown in Table S2 and S3. Figs. 4 and 5 showed correlations of non-crustal Fe with anthropogenic elements in coarse and fine aerosol particles, respectively. Therefore, their correlation matrixes were made separately. |
| 2.8. | Line 274: Please change "1.99 ± 0.892" to "1.99 ± 0.89". Many similar issues in the manuscript. Line 381 and 386: The number of decimal places should be consistent throughout the manuscript. | Thank you for pointing out. We have checked the entire manuscript and corrected these problems. |
| 2.9. | Line 311 missing recent reference npj Climate and Atmospheric Science 5(1), 53. | Thank you for your suggestion. We have added the reference of Liu et al. (2022). |
| 2.10. | Line 368: Remove "that" in the statement of "One of the reasons is that that……" | We follow the suggestion. We corrected the sentence as described below: One of the reasons is that the [d-Fe]/[d-Al] ratio in mineral dust differs from that in non-crustal sources, as will be discussed below (Fig. 6a). |
| 2.11. | Line 690: The descriptions off (g), (h) and (i) are not consistent with Figure 2 (g) and (h). | Thank you for pointing this out and we apologize for the inconvenience during your peer-preview processes. We have revised the caption of the figure. In addition, Figure 2 was moved to Figures S3 and S4 in Supplemental Information to reduce figures in the manuscript and to improve the resolution of the image. |
| 2.12. | Line 705: Figure 3c is the enrichment factor of Fe, not the Fesol% in each size fraction. | Thank you for pointing this out. We have revised it. |